# Soil carbon-concentration and carbon-climate feedbacks in CMIP6 Earth system models

Rebecca M. Varney[1, 2], Pierre Friedlingstein[1, 3], Sarah E. Chadburn[1], Eleanor J. Burke[4], and Peter M. Cox[1, 2]

[1]Faculty of Environment, Science and Economy, University of Exeter, Laver Building, North Park Road, Exeter, EX4 4QE, UK
[2]Global Systems Institute, University of Exeter, Laver Building, North Park Road, Exeter, EX4 4QE, UK
[3]Laboratoire de Météorologie Dynamique/Institut Pierre-Simon Laplace, CNRS, Ecole Normale Supérieure/Université PSL, Sorbonne Université, Ecole Polytechnique, Paris, 75231, France
[4]Met Office Hadley Centre, FitzRoy Road, Exeter, EX1 3PB, UK

**Correspondence:** Rebecca M. Varney (r.varney@exeter.ac.uk)

**Abstract.** Achieving climate targets requires mitigation against climate change, but also understanding of the response of land and ocean carbon systems. In this context, global soil carbon stocks and its response to environmental changes is key. This paper quantifies the global soil carbon feedbacks due to changes in atmospheric $CO_2$, and associated climate changes, for Earth system models (ESMs) in CMIP6. A standard approach is used to calculate carbon cycle feedbacks, defined here as soil carbon-concentration ($\beta_s$) and carbon-climate ($\gamma_s$) feedback parameters, which are also broken down into processes which drive soil carbon change. The sensitivity to $CO_2$ is shown to dominate soil carbon changes at least up to a doubling of atmospheric $CO_2$. However, the sensitivity of soil carbon to climate change is found to become an increasingly important source of uncertainty under higher atmospheric $CO_2$ concentrations.

## 1 Introduction

Global soil carbon stocks contain at least twice as much carbon than is stored in the world's vegetation, making soils the largest active store of carbon on the land surface of Earth (Canadell et al., 2021). In the absence of human disturbance and land-use change (Jones et al., 2018), future changes in soil carbon depend on the sensitivity to increases in atmospheric $CO_2$ concentrations and the sensitivity to the associated impacts, such as increases to atmospheric temperatures and changes in precipitation patterns (Varney et al., 2023; Todd-Brown et al., 2014). The quantification of such carbon cycle feedbacks is required to determine the overall response of the climate system to given anthropogenic $CO_2$ emissions and to help achieve Paris Agreement targets (Friedlingstein et al., 2022; Gregory et al., 2009).

Previous studies have defined land carbon cycle feedbacks within Earth system models (ESMs) from both CMIP6 and CMIP5 ensembles (Arora et al., 2020, 2013). In general, the overall response of carbon stores is separated into those due to changes in atmospheric $CO_2$ ($\Delta CO_2$) and those due to changes in global temperature ($\Delta T$), with the latter assumed to represent the overall impacts of climate change on large spatial scales. These components of land carbon cycle feedbacks are called carbon-concentration feedbacks ($\beta_L$) and carbon-climate feedbacks ($\gamma_L$), respectively (Friedlingstein et al., 2003, 2006). An

advantage of using this formulation is that it allows for the quantification of the feedbacks for a given atmospheric $CO_2$ concentration, which can then be used as a simplified measure to compare amongst ESMs despite the increasing model complexities (Arora et al., 2020, 2013; Gregory et al., 2009). For example, it provides a consistent metric to measure land carbon feedbacks

despite the differing climate sensitivities amongst ESMs (Boer and Arora, 2013).

In this study, soil carbon driven feedbacks in ESMs are quantified using this $\beta\gamma$ formulation (Friedlingstein et al., 2006). Additionally, the $\beta\gamma$ formulation is combined with the Varney et al. (2023) framework, which breakdowns future changes in soil carbon ($\Delta C_s$) into individual processes which drive this response. This paper makes use of the latest generation of the Coupled Model Intercomparison Project (CMIP6) used within the Intergovernmental Panel on Climate Change 6[th] Assessment Report

(IPCC AR6; IPCC (2021); Eyring et al. (2016)). To do this, soil carbon-concentration and carbon-climate feedback parameters are presented for CMIP6 ESMs, named $\beta_s$ and $\gamma_s$ respectively, together with components which make up $\beta_s$ and $\gamma_s$ due to associated processes. The aim of this paper is to: (1) quantify the sensitivity of soil carbon to increased atmospheric $CO_2$ concentrations and associated climate impacts by calculating $\beta_s$ and $\gamma_s$ for CMIP6 ESMs; (2) investigate the linearity of future soil carbon change at higher levels of atmospheric $CO_2$ increase; and (3) identify the fraction of the land carbon response to

climate change that is due to global soils.

## 2   Methods

### 2.1   C4MIP simulations

The Coupled Climate-Carbon Cycle Model Intercomparison Project (C4MIP) was set up to provide a common framework to allow for comparison and consistent evaluation of carbon cycle feedbacks within ESMs (Friedlingstein et al., 2006) and

has been used across CMIP generations (Arora et al., 2013, 2020). This framework includes a set of idealised experiments to simplify and quantify the impact of increasing atmospheric $CO_2$ on the climate system. In these experiments, additional effects such as land-use change, aerosols and non-$CO_2$ greenhouse gases are not included and nitrogen deposition is fixed at pre-industrial values (Jones et al., 2016).

The control simulation is known as the 1% $CO_2$ run (CMIP simulation *1pctCO2*), where a consistent 1% increase in at-

mospheric $CO_2$ per year is prescribed (referred to in this study as the full 1% $CO_2$ simulation), starting from pre-industrial concentrations and running for 150 years. Additional experiments were designed to enable the $CO_2$ and climate effects to be isolated, these are known as: biogeochemically coupled (referred to here as the 'BGC' simulation) and radiatively coupled (referred to here as the 'RAD' simulation) runs. In the BGC runs (CMIP6 simulation *1pctCO2-bgc* and CMIP5 simulation *esmFixClim1*), the 1% $CO_2$ increase per year only affects the carbon cycle component of the ESM while the radiation code

continues to see pre-industrial $CO_2$ values. Conversely, in the RAD runs (CMIP6 simulation *1pctCO2-rad* and CMIP5 simulation *esmFdbk1*), the 1% $CO_2$ increase per year affects only the radiation code, and the carbon cycle component of the ESM continues to see just the pre-industrial $CO_2$ value (285 ppm).

This study uses the full 1% $CO_2$, BGC, and RAD C4MIP experiments with 10 CMIP6 ESMs (Eyring et al., 2016): ACCESS-ESM1-5, BCC-CSM2-MR, CanESM5, CESM2, GFDL-ESM4, IPSL-CM6A-LR, MIROC-ES2L, MPI-ESM1-2-

LR, NorESM2-LM and UKESM1-0-LL (see Table 1). For comparison, the soil carbon feedback parameters were calcu-
lated using 6 CMIP5 ESMs (Taylor et al., 2012): CanESM2, GFDL-ESM2M, IPSL-CM5A-LR, MPI-ESM-LR, NorESM1-
ME and HadGEM2-ES (see Table A2). The ESMs included were chosen due to the availability of the data required at the
time of analysis (CMIP6: https://esgf-node.llnl.gov/search/cmip6/, last access: 4 February 2024, and CMIP5: https://esgf-
node.llnl.gov/search/cmip5/, last access: 6 February 2024).

## 2.2  Defining soil carbon feedbacks

### 2.2.1  Friedlingstein et al. (2006) $\beta\gamma$ formulation

The standard formulation uses a linear approximation to estimate carbon cycle feedbacks under a changing climate (Friedling-
stein et al., 2003, 2006). The change in land carbon storage ($\Delta C_L$, PgC) is approximated linearly using feedback parameters
which define separate sensitivities to changes in atmospheric $CO_2$ ($\Delta CO_2$, ppm) and changes in global temperatures ($\Delta T$,
°C), defined as the land carbon-concentration ($\beta_L$, PgC ppm$^{-1}$) and carbon-climate ($\gamma_L$, PgC °C$^{-1}$) (Equation 1).

$$\Delta C_L \approx \beta_L \Delta CO_2 + \gamma_L \Delta T \tag{1}$$

The Friedlingstein et al. (2006) methodology uses time-integrated fluxes, which represent the total change in size of the land
carbon pool ($\Delta C_L$). This is presented for the full 1% $CO_2$ simulation (Equation 2), BGC simulation (Equation 3), and RAD
simulation (Equation 4) below, where $\Delta C_L$, $\Delta C_L^{BGC}$, and $\Delta C_L^{RAD}$ are the changes in global land carbon pools (PgC), and
$F_L$, $F_L^{BGC}$, and $F_L^{RAD}$ are the net carbon fluxes to the land (PgC yr$^{-1}$), for each simulation.

$$\Delta C_L = \int F_L \, dt \approx \beta_L \Delta CO_2 + \gamma_L \Delta T \tag{2}$$

$$\Delta C_L^{BGC} = \int F_L^{BGC} \, dt \approx \beta_L \Delta CO_2 + \gamma_L \Delta T^{BGC} \approx \beta_L \Delta CO_2 \tag{3}$$

$$\Delta C_L^{RAD} = \int F_L^{RAD} \, dt \approx \gamma_L \Delta T^{RAD} \tag{4}$$

In these equations, $\Delta CO_2(t)$ (ppm) is consistent between all scenarios. Within the RAD simulation however (Equation
4), the carbon cycle does not see an increased $CO_2$ so the $\Delta CO_2$ is neglected and only found in the full 1% $CO_2$ and BGC
simulations (Equations 2 and 3, respectively). $\Delta T$, $\Delta T^{BGC}$, and $\Delta T^{RAD}$ (°C) are the changes in global temperatures, in
the full 1% $CO_2$, BGC, and RAD simulations, respectively. In Equation 3, $\Delta T^{BGC}$ is assumed to be negligible, following
Friedlingstein et al. (2006). As the increased $CO_2$ within the BGC simulation does not affect the radiation code, there is no
direct increase in atmospheric temperatures within the model. Arora et al. (2020) explain however, that local changes in the

carbon cycle arising from increases in $CO_2$ affect latent and sensible heat fluxes at the land surface, including: changes to evaporative fluxes from stomatal closure over land and changes in vegetation structure and coverage if dynamic vegetation is included within the ESM (see Table 1). This study assumes that the global temperature changes in the BGC simulation are negligible in the context of the $\beta\gamma$ formulation (Fig. A1).

### 2.2.2 Soil carbon-concentration and carbon-climate feedbacks

Global $\Delta C_L$ can be written as the sum of the changes in vegetation carbon ($\Delta C_v$) and changes in soil carbon ($\Delta C_s$). Following the $\beta\gamma$ formulation, a similar breakdown of the land carbon-concentration and carbon-climate feedback parameters can be derived, where $\beta_L = \beta_v + \beta_s$ and $\gamma_L = \gamma_v + \gamma_s$ (Equation 5).

$$\Delta C_L \approx (\beta_v + \beta_s)\Delta CO_2 + (\gamma_v + \gamma_s)\Delta T \tag{5}$$

$$\Delta C_v \approx \beta_v \Delta CO_2 + \gamma_v \Delta T \tag{6}$$

$$\Delta C_s \approx \beta_s \Delta CO_2 + \gamma_s \Delta T \tag{7}$$

Therefore, an equation for $\Delta C_s$ can be obtained, with soil specific carbon-concentration ($\beta_s$) and carbon-climate ($\gamma_s$) feedback parameters, which represent the sensitivity of $\Delta C_s$ to $CO_2$ and T, respectively (Equation 7).

### 2.3 Processes driving soil carbon change and relation to the $\beta\gamma$ formulation

To isolate the processes which make up each soil carbon feedback, we follow the framework presented in Varney et al. (2023). An equation for soil carbon (Equation 8) is derived using the definition of soil carbon turnover time ($\tau_s = C_s/R_h$), which is defined as the ratio of soil carbon storage ($C_s$) to the carbon output flux from the soil (heterotrophic respiration, $R_h$; Varney et al. (2020)). Future soil carbon can then be defined as initial soil carbon ($C_{s,0}$) plus a change in soil carbon ($\Delta C_s$), as shown by Equation 9, where the subscript 0 denotes the initial state (decadal time-average at the start of C4MIP simulation). Equation 9 can be expanded to give Equation 10, which can be simplified to give Equation 11, as shown below.

$$C_s = R_h \tau_s \tag{8}$$

$$C_{s,0} + \Delta C_s = (R_{h,0} + \Delta R_h)(\tau_{s,0} + \Delta \tau_s) \tag{9}$$

$$C_{s,0} + \Delta C_s = R_{h,0}\tau_{s,0} + \tau_{s,0}\Delta R_h + R_{h,0}\Delta\tau_s + \Delta R_h\Delta\tau_s \tag{10}$$

$$\Delta C_s = \tau_{s,0}\Delta R_h + R_{h,0}\Delta\tau_s + \Delta R_h\Delta\tau_s \tag{11}$$

To consider the above and below ground effects on soil carbon separately, the effects due to changes vegetation productivity,
represented by Net Primary Productivity (NPP), and effects due to changes in soil carbon turnover time due to increased heterotrophic respiration ($\tau_s$), are considered (Todd-Brown et al., 2014). However, due to the difference between the global fluxes NPP and $R_h$ in a transient climate, an additional term is included which is defined as Net Ecosystem Productivity ($NEP = NPP - R_h$). Using the definition of NEP, this can be substituted into Equation 11 to give Equation 12, and expanded to give an equation for $\Delta C_s$ in terms of NPP, NEP and $\tau_s$ (Equation 13, where the subscript 0 denotes the initial state).

$$\Delta C_s = \tau_{s,0}\Delta(NPP - NEP) + (NPP_0 - NEP_0)\Delta\tau_s + \Delta(NPP - NEP)\Delta\tau_s \tag{12}$$

$$\Delta C_s = \tau_{s,0}\Delta NPP + NPP_0\Delta\tau_s + \Delta NPP\Delta\tau_s - \tau_{s,0}\Delta NEP - NEP_0\Delta\tau_s - \Delta NEP\Delta\tau_s \tag{13}$$

The individual terms in Equation 13 are: the change in soil carbon due to NPP changes ($\Delta C_{s,NPP} \approx \tau_{s,0}\Delta NPP$), the change in soil carbon due to the NEP transient term ($\Delta C_{s,NEP} \approx -\tau_{s,0}\Delta NEP$), the change in soil carbon due to $\tau_s$ changes ($\Delta C_{s,\tau} \approx NPP_0\Delta\tau_s$), as well as the transient effect on $\tau_s$ ($\Delta C_{s,\tau_{NEP}} \approx -NEP_0\Delta\tau_s$). The two additional terms are the
non-linear term between NPP and $\tau_s$ ($\Delta NPP\Delta\tau_s$) and the non-linear term between NEP and $\tau_s$ ($\Delta NEP\Delta\tau_s$).

The NEP term is used to represent the transient state of the system where NPP $\neq R_h$. However, it is noted that if the initial state is in equilibrium, then the initial NEP ($NEP_0$) will be approximately equal to zero. This would mean the $\Delta C_{s,\tau_{NEP}}$ term (representing the difference in $\tau_s$ from using NPP or $R_h$ in the definition) will be negligible. Despite initialising at the start of the C4MIP simulations (decadal time-average at the start of C4MIP simulation), this term is included within the analysis for
completeness to ensure exact values of $\Delta C_s$.

Following on from this Varney et al. (2023) framework, the equation for $\Delta C_s$ (Equation 13) can also be defined for the change in soil carbon in both the BGC simulations ($\Delta C_s^{BGC}$, Equation 14) and RAD simulations ($\Delta C_s^{RAD}$, Equation 15), where the superscripts denotes the BGC and RAD simulations, respectively.

$$\Delta C_s^{BGC} = \tau_{s,0}^{BGC}\Delta NPP^{BGC} + NPP_0^{BGC}\Delta\tau_s^{BGC} + \Delta NPP^{BGC}\Delta\tau_s^{BGC}$$
$$- \tau_{s,0}^{BGC}\Delta NEP^{BGC} - NEP_0^{BGC}\Delta\tau_s^{BGC} - \Delta NEP^{BGC}\Delta\tau_s^{BGC} \tag{14}$$

$$\Delta C_s^{RAD} = \tau_{s,0}^{RAD}\Delta NPP^{RAD} + NPP_0^{RAD}\Delta\tau_s^{RAD} + \Delta NPP^{RAD}\Delta\tau_s^{RAD}$$
$$- \tau_{s,0}^{RAD}\Delta NEP^{RAD} - NEP_0^{RAD}\Delta\tau_s^{RAD} - \Delta NEP^{RAD}\Delta\tau_s^{RAD} \tag{15}$$

These equations can be used to investigate the sensitivity of these isolated processes to changes in atmospheric $CO_2$ and global temperature (T), as shown by Equations 16 and 17. This is done by the explicit differentiation of Equations 14 and 15 with respect to $CO_2$ and $T$, respectively.

$$\Delta C_s^{BGC} = \frac{\partial}{\partial CO_2}\left[\Delta C_s^{BGC}\right]\Delta CO_2 \tag{16}$$

$$\Delta C_s^{RAD} = \frac{\partial}{\partial T}\left[\Delta C_s^{RAD}\right]\Delta T \tag{17}$$

Equations 16 and 17 can be used to relate these $CO_2$ and T sensitivities to the $\beta\gamma$ formulation, where $\beta$ is used to represent the sensitivity to $CO_2$ and $\gamma$ is used to represent the sensitivity to T. Equation 7 which defines $\Delta C_s$ in terms of the soil carbon-concentration ($\beta_s$) and carbon-climate ($\gamma_s$) feedback parameters can be rewritten in terms of partial derivatives, as shown by Equation 18.

$$\Delta C_s = \frac{\partial C_s}{\partial CO_2}\Delta CO_2 + \frac{\partial C_s}{\partial T}\Delta T, \quad where, \beta_s = \partial C_s/\partial CO_2 \ and \ \gamma_s = \partial C_s/\partial T. \tag{18}$$

Then, Equations 16 and 17 can be used together with Equation 18 to combine the $\beta\gamma$ formulation with the Varney et al. (2023) framework. In this case, therefore $\beta_s$ and $\gamma_s$ can be defined as the contributions to $\Delta C_s$ based on the individual sensitivities of the soil carbon controls to $CO_2$ and T (by substituting Equations 14 and 15 into Equations 16 and 17, respectively), as shown by Equations 20 and 21.

$$\Delta C_s = \frac{\partial}{\partial CO_2}\left[\Delta C_s^{BGC}\right]\Delta CO_2 + \frac{\partial}{\partial T}\left[\Delta C_s^{RAD}\right]\Delta T \tag{19}$$

Where,

$$
\begin{aligned}
\beta_s = \tau_{s,0}^{BGC}\frac{\partial NPP^{BGC}}{\partial CO_2} &+ NPP_0^{BGC}\frac{\partial \tau_s^{BGC}}{\partial CO_2} + \frac{\partial \Delta NPP^{BGC}\Delta\tau_s^{BGC}}{\partial CO_2}\\
&- \tau_{s,0}^{BGC}\frac{\partial NEP^{BGC}}{\partial CO_2} - NEP_0^{BGC}\frac{\partial \tau_s^{BGC}}{\partial CO_2}\\
&- \frac{\partial \Delta NEP^{BGC}\Delta\tau_s^{BGC}}{\partial CO_2}
\end{aligned}
\tag{20}
$$

$$
\begin{aligned}
\gamma_s = \tau_{s,0}^{RAD}\frac{\partial NPP^{RAD}}{\partial T} &+ NPP_0^{RAD}\frac{\partial \tau_s^{RAD}}{\partial T} + \frac{\partial \Delta NPP^{RAD}\Delta\tau_s^{RAD}}{\partial T}\\
&- \tau_{,0s}^{RAD}\frac{\partial NEP^{RAD}}{\partial T} - NEP_0^{RAD}\frac{\partial \tau_s^{RAD}}{\partial T}\\
&- \frac{\partial \Delta NEP^{RAD}\Delta\tau_s^{RAD}}{\partial T}
\end{aligned}
\tag{21}
$$

Equations 20 and 21 can be rewritten by defining $\beta_s$ and $\gamma_s$ contribution terms, where each component of the equations make up the total $\beta_s$ and $\gamma_s$ sensitivities. As shown below for $\beta_s$ (Equation 22) and $\gamma_s$ (Equation 23).

$$\beta_s = \beta_{NPP} + \beta_{\tau} + \beta_{\Delta NPP \Delta \tau} - \beta_{NEP} - \beta_{NEP_{\tau}} - \beta_{\Delta NEP \Delta \tau} \tag{22}$$

$$\gamma_s = \gamma_{NPP} + \gamma_{\tau} + \gamma_{\Delta NPP \Delta \tau} - \gamma_{NEP} - \gamma_{NEP_{\tau}} - \gamma_{\Delta NEP \Delta \tau} \tag{23}$$

Where, $\beta_{NPP}$ and $\gamma_{NPP}$ are the $\beta\gamma$ contributions due to $\Delta$NPP, $\beta_{\tau}$ and $\gamma_{\tau}$ are the $\beta\gamma$ contributions due to $\Delta\tau_s$, $\beta_{NEP}$ and $\gamma_{NEP}$ are the $\beta\gamma$ contributions due to the transient NEP term, including $\beta_{NEP_{\tau}}$ and $\gamma_{NEP_{\tau}}$ representing the $\beta\gamma$ contributions due to the transient NEP term on $\Delta\tau_s$, and then $\beta_{\Delta NPP \Delta \tau}$, $\beta_{\Delta NEP \Delta \tau}$, $\gamma_{\Delta NPP \Delta \tau}$ and $\gamma_{\Delta NEP \Delta \tau}$ are the non-linear effects on $\beta\gamma$.

## 2.4 Calculation of feedback parameters

### 2.4.1 Defining climate variables

For each of the CMIP6 ESMs, the CMIP output variables: *cSoil*, *cLitter*, and *cVeg* are considered in the land carbon storage analysis. Soil carbon ($C_s$) is defined as the sum of carbon stored in soils and the carbon stored in the litter (CMIP variable *cSoil* + CMIP variable *cLitter*), allowing for a more consistent comparison between the models despite differences in how soil carbon and litter carbon are simulated (Varney et al., 2022; Todd-Brown et al., 2013). For models that do not report a separate litter carbon pool (CMIP variable *cLitter*), soil carbon is taken to be simply the CMIP variable *cSoil* (UKESM1-0-LL). Land carbon ($C_L$) is defined as the sum of carbon stored in soil + litter ($C_s$), plus the carbon stored in vegetation ($C_v$, CMIP variable *cVeg*). Global total values for $C_s$ and $C_L$ (PgC) are calculated using an area weighted sum (using the model land surface fraction, CMIP variable *sftlf*).

In the breakdown analysis of the $\beta\gamma$ feedbacks, Net Primary Productivity (NPP, CMIP variable *npp*) is defined as the net carbon assimilated by plants via photosynthesis minus loss due to plant respiration and is used to represent the net carbon input flux to the system. Heterotrophic Respiration ($R_h$, CMIP variable *rh*) is defined as the microbial respiration within global soils and is used to define an effective global soil carbon turnover time ($\tau_s$). $\tau_s$ (years) is defined as the ratio of mean soil carbon to annual mean heterotrophic respiration, given as $\tau_s = C_s/R_h$ (where the mean represents an area weighted global average). Carbon fluxes (NPP and $R_h$) in the calculation of feedback contributions are considered as area weighted global totals in units of PgC yr$^{-1}$ (using the model land surface fraction, CMIP variable *sftlf*).

Increases in global temperatures ($\Delta T$) are considered using CMIP variable *tas*, which is defined as the change in near-surface air temperature (°C). To calculate changes in atmospheric $CO_2$ ($\Delta CO_2$) in the C4MIP 1% $CO_2$ simulations, initial pre-industrial $CO_2$ concentrations are assumed to be 285 ppm, and then cumulatively increased by 1% each year, for 70 years (approximately 2x$CO_2$) or 140 years (approximately 4x$CO_2$).

### 2.4.2 Carbon-concentration feedback parameter ($\beta$)

To calculate the soil carbon-concentration feedback parameter ($\beta_s$), the BGC run was used. For each ESM, the change in soil carbon in the BGC run ($\Delta C_s^{BGC}$, PgC) was divided by the change in $CO_2$ concentration (ppm) up to that point in time (expressed in units of carbon uptake or release per unit change in $CO_2$, PgC ppm$^{-1}$). For this study, $\beta_s$ was calculated at the time of 2x$CO_2$ and 4x$CO_2$. To calculate the land carbon-concentration feedback parameter ($\beta_L$), the same method was used but replacing $C_s^{BGC}$ with $C_L^{BGC}$.

### 2.4.3 Carbon-climate feedback parameter ($\gamma$)

To calculate the soil carbon-climate feedback parameter ($\gamma_s$), the RAD run was used. For each ESM, the change in soil carbon in the RAD run ($\Delta C_s^{RAD}$, PgC) was divided by the change in temperature T (°C) up to that point in time (expressed in units of carbon uptake or release per unit change in temperature, PgC °C$^{-1}$). For this study, $\gamma_s$ was calculated at 2x$CO_2$ and 4x$CO_2$. To calculate the land carbon-climate feedback parameter ($\gamma_L$), the same method was used but replacing $C_s^{RAD}$ with $C_L^{RAD}$.

### 2.4.4 Feedback parameter contributions

To calculate the isolated contributions which make up $\beta$ and $\gamma$, as shown in Equations 22 and 23, again the BGC and RAD simulations are used for each CMIP6 ESM. To calculate gradients with respect to $CO_2$ and T, the methodology presented above is used, but with the relevant component against $CO_2$ or T, such as NPP or $\tau_s$. The $\beta_s$ contributions are expressed in units of carbon uptake or release per unit change in $CO_2$ (PgC ppm$^{-1}$) and the $\gamma_s$ contributions are expressed in units of carbon uptake or release per unit change in temperature (PgC °C$^{-1}$), using the definitions presented in Equations 22 and 23.

## 3 Results

### 3.1 Projections of soil carbon change

Projections of soil carbon change in CMIP6 ESMs for the full 1% $CO_2$ ($\Delta C_s$), BGC ($\Delta C_s^{BGC}$) and RAD ($\Delta C_s^{RAD}$) simulations are presented in Fig. 1. Soil carbon is projected to increase in the full 1% $CO_2$ simulation amongst CMIP6 ESMs (ensemble mean 88.2 $\pm$ 40.4 PgC at 2x$CO_2$ and 177 $\pm$ 141 PgC at 4x$CO_2$). However, the magnitude of the increase varies amongst the ESMs, with a range of 38 PgC (NorESM2-LM) to 145 PgC (BCC-CSM2-MR) at 2x$CO_2$, and a range of 15 PgC (ACCESS-ESM1-5) to 502 PgC (CanESM5) at 4x$CO_2$. Six of the ESMs (CanESM5, CESM2, GFDL-ESM4, MIROC-ES2L, MPI-ESM1-2-LR, NorESM2-LM) see an increased $\Delta C_s$ value with increasing climate forcing, however the remaining four ESMs (ACCESS-ESM1-5, BCC-CSM2-MR, IPSL-CM6A-LR, UKESM1-0-LL) see a saturation to the rate of increase, or even a turning point where carbon starts to decrease again, from 70 years ($\approx$ 2x$CO_2$) in the simulation (Fig. 1(a)).

The projected increase in soil carbon can be approximated by the increases projected in the BGC run ($\Delta C_s^{BGC}$; ensemble mean 132 $\pm$ 66.5 PgC at 2x$CO_2$ and 348 $\pm$ 203 PgC at 4x$CO_2$, Fig. 1(b)) and the decreases projected in the RAD run ($\Delta C_s^{RAD}$; ensemble mean -45.5 $\pm$ 22.9 PgC at 2x$CO_2$ and -170 $\pm$ 94.7 PgC at 4x$CO_2$, Fig. 1(c)). The response due to

increases in atmospheric $CO_2$ (BGC simulation) are found to dominate the overall response (full 1% $CO_2$ simulation) in the majority of models, where greater magnitudes of change are seen compared to the RAD simulation (exception ACCESS-ESM1-5). The BGC simulation also sees a greater spread in projected $\Delta C_s$, with a range of 218 PgC at 2x$CO_2$ and 603 PgC at 4x$CO_2$ ($\Delta C_s^{BGC}$), compared to ranges of 68 PgC at 2x$CO_2$ and 312 PgC at 4x$CO_2$ in the RAD simulation ($\Delta C_s^{RAD}$).

Fig. 2 shows patterns of soil carbon changes at 4x$CO_2$ for the full 1% $CO_2$ ($\Delta C_s$), BGC ($\Delta C_s^{BGC}$) and RAD ($\Delta C_s^{RAD}$). In the BGC simulation, increases in $\Delta C_s^{BGC}$ are seen across the majority of regions within CMIP6 ESMs, though exceptions are found in the northern latitudes for two ESMs (CanESM5 and NorESM2-LM). Across the ensemble, the projected increases in $\Delta C_s^{BGC}$ have spatially varying magnitudes, where generally the greatest increases are seen in the tropical regions. Conversely, the RAD simulation generally sees reductions in $\Delta C_s^{RAD}$ globally, with the greatest reductions seen in the tropical regions. However, disagreement is seen in the northern latitudes, where four models (ACCESS-ESM1-5, CanESM5, MIROC-ES2L, UKESM1-0-LL) see an increased $\Delta C_s^{RAD}$ and three models (BCC-CSM2-MR, CESM2, NorESM2-LM) see a decreased $\Delta C_s^{RAD}$. The overall $\Delta C_s$ values seen in the full 1% $CO_2$ simulation are again found to be mostly dominated by the BGC simulation (Fig. 2), though exceptions are seen where the RAD simulation is shown to dominate the response for certain regions. Specifically, the reduced $\Delta C_s$ within the RAD simulation dominates the net response in the northern latitudes of three ESMs (BCC-CSM2-MR, CESM2, and NorESM2-LM; the only models where decreases where seen), as well as in the tropical regions of a different three ESMs (ACCESS-ESM1-5, GFDL-ESM4, and UKESM1-0-LL).

### 3.2 Soil carbon-concentration and carbon-climate feedback parameters

The calculated $\beta_s$ and $\gamma_s$ values for CMIP6 ESMs are presented in Table 2. Values for $\beta_s$ are found to be positive amongst the CMIP6 ESMs which is consistent with increased $C_s$ with increasing $CO_2$, and values for $\gamma_s$ are found to be negative which is consistent with decreased $C_s$ with increasing temperature (Fig. 3). The magnitude of the feedback parameters ($\beta_s$ and $\gamma_s$) are found to vary amongst the CMIP6 ensemble, suggesting uncertainty in the magnitude of the soil carbon response to climate change. Generally, models with higher sensitivities to $CO_2$ ($\beta_s$), also have higher sensitivities to temperature ($\gamma_s$), where a $r^2$ values of 0.64 (2x$CO_2$) and 0.60 (4x$CO_2$) are found between the $\beta_s$ and $\gamma_s$ values (Table 2). The range in projected $\beta_s$ parameters are found to be relatively consistent between 2x$CO_2$ and 4x$CO_2$ (where a small decrease is seen), with a range of 0.704 PgC ppm$^{-1}$ and range of 0.636 PgC ppm$^{-1}$ respectively. Conversely, the range of calculated $\gamma_s$ parameters are found to be less consistent between 2x$CO_2$ and 4x$CO_2$ (increasing range with increased global warming), with ranges of 42.7 PgC °C$^{-1}$ and 68.0 PgC °C$^{-1}$ respectively (Table 2).

The linearity of future soil carbon changes can be investigated by comparing the 2x$CO_2$ and 4x$CO_2$ lines for $\beta_s$ and $\gamma_s$ in Fig. 3. A future linear response is shown to be a good approximation, however the figure suggests a slight non-linearity in the soil carbon response to both $CO_2$ ($\Delta C_s^{BGC}$) and temperature ($\Delta C_s^{RAD}$) in the majority of ESMs. The BGC simulation generally sees greater consistency between 2x$CO_2$ and 4x$CO_2$ $\beta_s$ values, for example in the CESM2 and NorESM2-LM models. However, the majority of ESMs (ACCESS-ESM1-5, BCC-CSM2-MR, GFDL-ESM4, IPSL-CM6A-LR, MIROC-ES2L, MPI-ESM1-2-LR, and UKESM1-0-LL) see a reduction in $\beta_s$ and a saturation to the sensitivity with greater $CO_2$ levels (Fig. 3(a)). In the RAD simulation, generally inconsistencies are seen between 2x$CO_2$ and 4x$CO_2$ (exception MPI-ESM1-2-LR) and an

increased sensitivity of $C_s^{RAD}$ to temperature (T) with increased climate forcing is suggested by the majority of CMIP6 ESMs (Fig. 3(b)). As an example, in CESM2 where one of the lowest sensitivities to T at 2xCO$_2$ is seen, the ESM see an approximate 50% increase in $\gamma_s$ by 4xCO$_2$ (Table 2).

The $\beta_s$ and $\gamma_s$ values were also calculated for CMIP5 ESMs (Table A3), which can be compared with a subset of generationally related CMIP6 ESMs considered in this study (Fig. A3). The CMIP6 ensemble means for both $\beta_s$ and $\gamma_s$ parameters are found to be lower compared with the CMIP5 ensemble means (Table 2 and Table A3). The relationship of $\beta_s$ and $\gamma_s$ values between CMIP5 and CMIP6 however, is not found to be consistent amongst the ensembles. For $\beta_s$, reductions are seen in 4 ESMs (GFDL-ESM2M Vs GFDL-ESM4, IPSL-CM5A-LR Vs IPSL-CM6A-LR, MPI-ESM-LR Vs MPI-ESM1-2-LR, and

HadGEM2-ES Vs UKESM1-0-LL), compared to increases in the remaining 2 (CanESM2 Vs CanESM5 and NorESM1-ME Vs NorESM2-LM). For $\gamma_s$, a greater value (closer to 0) is seen in 4 ESMs (CanESM2 Vs CanESM5, GFDL-ESM2M Vs GFDL-ESM4, IPSL-CM5A-LR Vs IPSL-CM6A-LR, and MPI-ESM-LR Vs MPI-ESM1-2-LR), compared to a lower value (greater negative) is seen in the remaining 2 ESMs (NorESM1-ME Vs NorESM2-LM and HadGEM2-ES Vs UKESM1-0-LL).

### 3.3   Breakdown of the feedback parameters into soil carbon drivers

In this section, $\beta_s$ and $\gamma_s$ are broken down into the individual sensitivities of drivers of soil carbon change which make up the net response. As shown in Fig. 4, the total soil carbon sensitivities ($\beta_s$ and $\gamma_s$, blue bars) can be considered as a sum of the sensitivity due to $\Delta$NPP ($\beta_{NPP}$ and $\gamma_{NPP}$, green bars), the sensitivity due to $\Delta\tau_s$ ($\beta_\tau$ and $\gamma_\tau$, red bars), and additional terms due to the transient land carbon sink, such as NEP ($\beta_{NEP}$ and $\gamma_{NEP}$, light green bars) and the NEP effect on $\tau_s$ ($\beta_{\tau_{NEP}}$ and $\gamma_{\tau_{NEP}}$, pink bars). Additionally, there are non-negligible contributions due to non-linear sensitivities between NPP and $\tau_s$

($\beta_{\Delta NPP\Delta\tau}$ and $\gamma_{\Delta NPP\Delta\tau}$, black bars) and a small contribution from non-linear sensitivities between NEP and $\tau_s$ ($\beta_{\Delta NEP\Delta\tau}$ and $\gamma_{\Delta NEP\Delta\tau}$, grey bars).

Investigating the sensitivity of soil carbon to $\Delta$NPP, $\beta_{NPP}$ is found to be positive amongst CMIP6 ESMs (Fig. 4). At 2xCO$_2$, $\beta_{NPP}$ ranges from 0.567 PgC ppm$^{-1}$ (ACCESS-ESM1-5) to 5.62 PgC ppm$^{-1}$ (BCC-CSM2-MR), with an ensemble mean of 2.37 $\pm$ 1.37 PgC ppm$^{-1}$. There is some evidence of a saturation of global NPP at higher CO$_2$, with the sensitivity

of NPP to CO$_2$ ($\beta_{NPP}$) decreasing at 4xCO$_2$ to an ensemble mean of 1.44 $\pm$ 0.933 PgC ppm$^{-1}$. The sensitivity of NPP to global temperature changes ($\gamma_{NPP}$) is found to be more variable amongst the ensemble. The majority of models find $\gamma_{NPP}$ to be negative, however it is found to be positive in two ESMs (CanESM5 and MPI-ESM1-2-LR). The sensitivity of NPP to temperature ($\gamma_{NPP}$) is found to be more consistent with climate change than the sensitivity to CO$_2$ ($\beta_{NPP}$), where the $\gamma_{NPP}$ ensemble mean changes from -29.4 $\pm$ 40.1 PgC $^\circ$C$^{-1}$ at 2xCO$_2$ to -35.3 $\pm$ 33.1 PgC $^\circ$C$^{-1}$ at 4xCO$_2$ (Fig. 4). At 4xCO$_2$, the

lowest sensitivity of NPP to temperature is seen in CanESM5 (3.95 PgC $^\circ$C$^{-1}$), and the greatest sensitivity in BCC-CSM2-MR (-90.8 PgC $^\circ$C$^{-1}$).

Investigating the sensitivity of soil carbon to $\Delta\tau_s$, negative $\beta_\tau$ and $\gamma_\tau$ values are mostly found amongst the CMIP6 models (Fig. 4). An anomaly is found where $\tau_s$ is found to increase with temperature in the ACCESS-ESM1-5 model, where the reason for this is unclear (Fig. A2). The sensitivity of $\tau_s$ to T ($\gamma_\tau$) is also found to be more consistent with increasing climate change

than the sensitivity to CO$_2$, where an ensemble mean of -25.2 $\pm$ 27.9 PgC $^\circ$C$^{-1}$ at 2xCO$_2$ and -20.5 $\pm$ 29.5 PgC $^\circ$C$^{-1}$ at

4xCO$_2$ is seen. At 4xCO$_2$, the greatest sensitivity of $\tau_s$ to temperature is seen in the MIROC-ES2L model (-54.6 PgC $^\circ$C$^{-1}$) and the lowest sensitivity is seen in the NorESM2-LM model (-2.80 PgC $^\circ$C$^{-1}$). $\tau_s$ is found to decrease non-linearly with increasing CO$_2$ ($\beta_\tau$). At 2xCO$_2$, $\beta_\tau$ ranges from -0.329 PgC ppm$^{-1}$ (ACCESS-ESM1-5) to -1.90 PgC ppm$^{-1}$ (BCC-CSM2-MR), with an ensemble mean of -0.900 $\pm$ 0.574 PgC ppm$^{-1}$. Due to the non-linearity, a reduced ensemble mean of -0.450 $\pm$ 0.359 PgC ppm$^{-1}$ is found at 4xCO$_2$ compared with 2xCO$_2$ (Fig. 4).

It is apparent from Fig. 4 that the sensitivities of NPP and $\tau_s$ to both CO$_2$ and T must be accounted for to understand and quantify the sensitivities of soil carbon. The magnitude of $\beta_\tau$ is found to be approximately a third of the magnitude of $\beta_{NPP}$ at both 2xCO$_2$ and 4xCO$_2$, but with counteracting signs of change. Models with the lowest $\beta_{NPP}$ sensitivities also see the lowest $\beta_\tau$ sensitivities (e.g. ACCESS-ESM1-5), and via versa. The magnitude of $\gamma_{NPP}$ is generally found to be greater across the ensemble compared with $\gamma_\tau$, however with a greater range of sensitivities. Additionally, the apparent sensitivity of soil carbon to CO$_2$ is less then the individual sensitivities of NPP and $\tau_s$, due to a cancellation effect from opposing signs, leading to a lower apparent $\beta_s$. The magnitudes of $\beta_{NPP}$ and $\beta_\tau$ are lower at 4xCO$_2$ than 2xCO$_2$, which means a reduced sensitivity of NPP and $\tau_s$ to CO$_2$ at greater levels of climate change, However, due to this cancellation effect the same reduced sensitivity is not seen in $\beta_s$. Conversely, a reduced sensitivity of NPP and $\tau_s$ to temperature is not suggested under increasing climate forcing. No clear relationship between $\gamma_{NPP}$ and $\gamma_\tau$ is seen amongst the CMIP6 ESMs (Fig. 4).

The contribution of the non-linearity between NPP and $\tau_s$ to the net soil carbon sensitivity is also investigated ($\beta_{\Delta NPP \Delta \tau}$ and $\gamma_{\Delta NPP \Delta \tau}$). Fig. 4 suggests that the non-linearity between NPP and $\tau_s$ is more robustly projected to result from increasing CO$_2$ ($\beta_s$), however non-linearities in $\gamma_s$ are also seen in the models which the greatest temperature sensitivities. The ensemble mean predicted $\beta_{\Delta NPP \Delta \tau}$ is found to be -0.462 $\pm$ 0.462 at 2xCO$_2$ and -0.463 $\pm$ 0.468 PgC ppm$^{-1}$ at 4xCO$_2$. As expected from Fig. 4, predicted $\gamma_{\Delta NPP \Delta \tau}$ is found to have a low sensitivity, where the ensemble means of -0.374 $\pm$ 3.12 at 2xCO$_2$ and -0.0478 $\pm$ 7.42 PgC $^\circ$C$^{-1}$ at 4xCO$_2$ are found. Additionally, the NEP terms ($\beta_{NEP}$ and $\gamma_{NEP}$) are shown to contribute to both CO$_2$ and T sensitivities (Fig. 4), due to the disequilibrium of land carbon changes under 1% increasing CO$_2$.

## 3.4 Investigating robustness of the $\Delta C_s$ approximation

Projections of $\Delta C_s$ in ESMs in the full 1% CO$_2$ simulation was compared with the estimated $\Delta C_s$ derived using Equation 7, which uses the derived $\beta_s$ and $\gamma_s$ feedback parameters together with model specific $\Delta T$ and estimates for $\Delta$CO$_2$ (Fig. 5). This investigates the approximation that changes in the full 1% CO$_2$ simulation is equal to the sum of changes in the BGC and RAD simulations. At 2xCO$_2$, the approximation is found to predict $\Delta C_s$ within 20% of the actual projected values in the 1% CO$_2$ simulation for 7 out of the 10 CMIP6 ESMs (BCC-CSM2-MR, CESM2, GFDL-ESM4, IPSL-CM6A-LR, MIROC-ES2L, MPI-ESM1-2-LR and UKESM1-0-LL). At 4xCO$_2$, the robustness of the assumption between the BGC and RAD simulations reduces for future changes in soil carbon. However, $\beta_s \Delta CO_2 + \gamma_s \Delta T$ is within 20% of the projected $\Delta C_s$ for 5 out of the 10 ESMs (GFDL-ESM4, IPSL-CM6A-LR, MIROC-ES2L, MPI-ESM1-2-LR and UKESM1-0-LL). The models where the approximation is the least consistent with projected $\Delta C_s$ are ACCESS-ESM1-5 and BCC-CSM2-MR, where at 4xCO$_2$ the greatest non-linearities are present between BGC and RAD simulations (Fig. 5).

### 3.5 Comparisons between soil and land feedback parameters

The contribution of the sensitivity of soil carbon stocks ($C_s$) to the total sensitivity of land carbon stocks ($C_L$) was investigated by comparing the $\beta$ and $\gamma$ feedback parameters for land (Table A1) and soil (Table 2), for both 2xCO$_2$ and 4xCO$_2$ in CMIP6 ESMs (Fig. 6). Here, the assumption from Equation 5 is followed that the land sensitivity is made up of the sum of the soil and vegetation responses. For the carbon-concentration feedback ($\beta$), the portion of the land sensitivity to CO$_2$ ($\beta_L$) that is due to global soils ($\beta_s$) ranges from 19% (NorESM2-LM) to 53% (BCC-CSM2-MR), with a mean of $38 \pm 11$ % seen across

the CMIP6 ESMs at 2xCO$_2$ (Fig. 6(a)). Similar proportions are found at 4xCO$_2$, ranging from 22% (NorESM2-LM) to 58% (MIROC-ES2-L), with a mean of $42 \pm 12$ % seen across the CMIP6 ESMs (Fig. 6(b)). The portion of $\beta_L$ due to $\beta_s$ is estimated to be close to half the total land response. For the carbon-climate feedback ($\gamma$), the portion of the land sensitivity to climate ($\gamma_L$) that is due to global soils ($\gamma_s$) ranges from approximately 42% (CESM2) to 147% (MPI-ESM1-2-LM), with a mean of 75 $\pm$ 30 % seen across the CMIP6 ESMs at 2xCO$_2$ (Fig. 6(a)), and at 4xCO$_2$ the ranges is from 48% (ACCESS-ESM1-5) to 157%

(MPI-ESM1-2-LM), with a mean of $75 \pm 31$ % seen across the CMIP6 ESMs (Fig. 6(b)). Therefore, the portion of $\gamma_L$ due to $\gamma_s$ is estimated to be the majority of the sensitivity, suggesting that soil dominates the response of land carbon to climate. Note that the MPI-ESM1-2-LR model sees a greater $\gamma_s$ value compared with $\gamma_L$, resulting in the percentage of the land response attributed to soil being greater than 100%. This suggests a positive $\gamma_v$ response in this model, meaning a predicted increased vegetation carbon globally with global warming.

## 4 Discussion

Quantifying the future sensitivity of global soil carbon stocks to anthropogenic CO$_2$ emissions and their role within future land carbon storage is vital in order to understand future changes in the Earth's climate system (Canadell et al., 2021). Global changes in soil carbon ($\Delta C_s$), in the absence of human disturbance and land-use change, will result from responses due to changes in atmospheric CO$_2$ and associated changes in global temperatures (T), which is used to represent climate effects on a

325 global scale. By separating the sensitivities due to increasing CO$_2$ and T, the idealised C4MIP ESM simulations allows for these effects on soil carbon to be examined individually and the use of the $\beta\gamma$ formulation allows these sensitivities to be quantified and compared for CMIP6 ESMs (Jones et al., 2016; Friedlingstein et al., 2006). Further, combining the $\beta\gamma$ formulation with the Varney et al. (2023) $\Delta C_s$ framework, allows us to isolate the sensitivities of soil carbon processes which influence $\beta_s$ and $\gamma_s$ within models.

Across CMIP6 ESMs, soil carbon is projected to increase in the BGC simulation ('CO$_2$ only') and decrease in the RAD simulation ('climate only'), consistent with projections of the overall land carbon response (Arora et al., 2020). The BGC simulation has been used to quantify the sensitivity of soil carbon to $\Delta$CO$_2$ ($\beta_s$), where positive $\beta_s$ values were defined due to the projected increase in soil carbon with increased atmospheric CO$_2$ (Fig. 1(b)). The positive $\beta_s$ has been shown here to mostly be a result of a positive $\beta_{NPP}$ term (Fig. 4), which represents the increased CO$_2$ fertilisation effect describing an

increased vegetation productivity under higher atmospheric CO$_2$ concentrations, which leads to an increased input of litter carbon into soil carbon pools (Schimel et al., 2015; Koven et al., 2015). A negative contribution of $\beta_\tau$ on $\beta_s$ is also shown (Fig.

4). Previously, Varney et al. (2023) presented a transient reduction in $\tau_s$ in CMIP6 ESMs due to an increased rate of carbon input into the soil (i.e. negative $\beta_\tau$ due to positive $\beta_{NPP}$); a phenomenon known as false priming (Koven et al., 2015). However, it can be seen that the magnitude of this effect is small compared to the $CO_2$ fertilisation effect across the ESMs ($\beta_\tau$ Vs $\beta_{NPP}$, Fig. 4). Despite agreement on a net increase in soil carbon stocks globally (positive $\beta_s$), this study highlights uncertainty on the projected magnitude of this sensitivity amongst the CMIP6 models, which is seen to be driven by uncertainties in $\beta_{NPP}$ (Fig. 4).

The RAD simulation has been used to quantify the sensitivity of soil carbon to changes in climate ($\Delta$T; $\gamma_s$), where negative $\gamma_s$ values were defined due to the projected decrease in soil carbon with global warming (Fig. 1(c)). The negative $\gamma_s$ term has been shown here to be a result of negative $\gamma_\tau$, and in many cases negative $\gamma_{NPP}$ (Fig. 4). The negative sensitivity of $\tau_s$ to global warming (negative $\gamma_\tau$) is known to be due to an increased rate of heterotrophic respiration ($R_h$) under warmer temperatures as a result of increased microbial activity (Varney et al., 2020; Crowther et al., 2016). The global sensitivity of NPP to climate changes ($\gamma_{NPP}$) is less certain where both negative and positive values are seen across the CMIP6 ESMs (Fig. 4). This is likely due to more spatially varying responses, where the resultant $\Delta C_s$ can be seen in Fig. 2. For example, increased temperatures in northern latitudes could result in the northward expansion of boreal forests (Pugh et al., 2018), which would increase forest productivity and subsequently carbon storage in these regions. However, future changes in precipitation patterns could lead to regions with reduced soil moisture, which would conversely lead to reduced vegetation productivity and carbon uptake (Green et al., 2019). The uncertainties associated with projected spatial changes ($\gamma_{NPP}$), together with the uncertainties in the magnitude of carbon turnover times within the soil ($\gamma_\tau$; Varney et al. (2020); Koven et al. (2017)), results in uncertainties in the sensitivity of soil carbon to climate changes ($\gamma_s$) amongst the CMIP6 models.

This paper highlights the importance of soils within the land carbon response to global warming (Fig. 6). Despite the $\Delta C_s$ sensitivity to $CO_2$ dominating net soil carbon changes ($\beta_s$), it could be argued that the significance of the $\Delta C_s$ climate sensitivity ($\gamma_s$) will increase under more extreme levels of climate change. This is suggested by both a projected saturation of $\beta_s$ and an increase in $\gamma_s$ between 2x$CO_2$ and 4x$CO_2$ shown in the CMIP6 ensemble means (Table 2). The saturation, or reduced rate of increase, in $\beta_s$ seen in CMIP6 is likely due to a limit of the $CO_2$ fertilisation effect, based on the reduced $\beta_{NPP}$ values between 2x$CO_2$ and 4x$CO_2$ (Fig. 4). The rate of $CO_2$ fertilisation in the future is expected to be limited by nutrient availability (Wieder et al., 2015), which in CMIP6 is now more explicitly represented by the inclusion of an interactive nitrogen cycle in multiple models (see Table 1). This implementation is expected to limit the increased productivity from $CO_2$ fertilisation within ESMs (Davies-Barnard et al., 2020), and has previously been found to lower the magnitude of the land feedback parameters (Arora et al., 2020). However, it is noted that warming within the soil could accelerate nutrient mineralisation, which could result in a liberation of nitrogen due to increased microbial breakdown of plant litter, alleviating the nutrient limitation in plants (Todd-Brown et al., 2014).

Unlike the $\beta_s$ parameter, the sensitivity of soil carbon to climate changes ($\gamma_s$) has been shown to increase with global warming amongst CMIP6. The greater $\gamma_s$ values at 4x$CO_2$ compared to 2x$CO_2$ found here implies an increased rate of soil carbon loss under increased amounts of global warming (Table 2). Additionally, it could be hypothesised that limitations within CMIP6 ESMs in the representation of soil carbon and related processes could lead to a potential underestimation of $\gamma_s$. In Fig.

2, reductions in soil carbon stocks within the high northern latitudes are only seen in 3 models for the full 1% $CO_2$ simulation (BCC-CSM2-MR, CESM2, and NorESM2-LM). Varney et al. (2022) find that these CMIP6 models represent quantities of northern latitude carbon stocks the most consistently with observational estimates, which could imply an increased likelihood

of soil carbon loss from the northern latitudes based on consistency with observations. It is noted however, that CESM2 and NorESM2-LM contain the same land surface model so are expected to show similar results (Lawrence et al., 2019). Furthermore, the majority of ESMs do not include explicit representation of permafrost carbon (Burke et al., 2020). Including permafrost within ESMs would result in increased quantities of carbon within the soil known to be especially sensitive to global warming (increased $\gamma_s$), which currently are not included in the calculation of these feedbacks (Schuur et al., 2015).

The $\beta\gamma$ formulation has many benefits in allowing the quantification and comparison of land and soil carbon feedbacks amongst ESMs. However, one limitation is due to $\Delta C_s$ not being consistently linear with increasing $CO_2$ and temperature (Fig. 3), so the parameter values depend on the point in time which they are calculated (for example, 2x$CO_2$ or 4x$CO_2$). This has been shown to be due to non-linearities in the processes driving soil carbon feedbacks (Fig. 4), such as the discussed saturation of the $CO_2$ fertilisation effect ($\beta_{NPP}$; Wang et al. (2020)), and additionally a known $Q_{10}$ dependence of heterotrophic (soil)

respiration to temperature ($\gamma_\tau$; Zhou et al. (2009)).

Non-linearities between $CO_2$ and T responses are also known and have previously been shown within ESMs in the future land carbon responses (Schwinger et al., 2014; Zickfeld et al., 2011; Gregory et al., 2009). Zickfeld et al. (2011) suggest that the non-linearity in the land response are due to significantly differing vegetation responses which depend on whether or not climate effects are combined with the $CO_2$ fertilisation effect; for example, forest dieback (Cox et al., 2004). However, this is

model dependent as not all models within CMIP6 simulate dynamic vegetation (Table 1). The spatial variations in the response of soil carbon to $CO_2$ and climate that are seen in Fig. 2 could also contribute to the non-linearity. For example, a different spatial pattern of soil carbon under elevated $CO_2$ could lead to a different overall temperature response, e.g. if more carbon is in the high latitudes where greater temperature changes are seen. Arora et al. (2020) find that climate responses in the BGC simulation account for a difference of 1% - 5% in the calculation of the feedbacks, suggesting a small but non-negligible effect

of climate in the BGC runs. This response was shown to be dependent on the representation of vegetation within the model, as with the non-linearities found in Zickfeld et al. (2011). Despite this, isolating and quantifying the key sensitivities with the $\beta\gamma$ method provides a useful benchmark for feedbacks within CMIP.

## 5   Conclusions

The Friedlingstein et al. (2006) methodology adapted in this study suggests that $\beta_s$ and $\gamma_s$ linearity is a valid assumption for

projected soil carbon changes in ESMs up until a doubling of $CO_2$. However, under more extreme levels of climate change, the results here suggest the need for the non-linearity in feedbacks to be further investigated. Soil carbon is found to have a greater impact on carbon-climate feedbacks than vegetation carbon responses, which means that the sensitivity of soil carbon to changes in global temperature is the dominant response of the land carbon cycle when considering climate effects. Therefore, further understanding and quantifying the sensitivity of global soils under global warming is necessary to quantify future

changes in the climate system. Moreover, the sensitivity of soil carbon to temperature increases with increasing climate forcing, suggesting that soil carbon is particularly important in the long-term land carbon response under extreme levels of global warming.

*Code availability.* Code is available on GitHub (https://github.com/rebeccamayvarney/CMIP6-soil-beta-gamma) and Zenodo (https://doi.org/10.5281/zenodo.10927091).

*Data availability.* The CMIP6 and CMIP5 data analysed during this study is available online (cmip6: https: //esgf-node.llnl.gov/search/cmip6/, cmip5: https: //esgf-node.llnl.gov/search/cmip5/).

*Author contributions.* RMV and PMC outlined the study. RMV completed the analysis and produced the figures. All the co-authors provided useful guidance on the study at various times and suggested edits to the draft manuscript.

*Competing interests.* The authors have declared no competing interests.

*Acknowledgements.* This research has been supported by the European Research Council, Climate–Carbon Interactions in the Current Century project (4C; grant no. 821003) (RMV, PMC and PF) and Emergent Constraints on Climate–Land feedbacks in the Earth System project (ECCLES; grant no. 742472) (RMV and PMC). SEC was supported by a Natural Environment Research Council independent research fellowship (grant no. NE/R015791/1). EJB was supported by the Joint UK BEIS/Defra Met Office Hadley Centre Climate Programme (grant no. GA01101). We thank the World Climate Research Programme's Working Group on Coupled Modelling and the climate modelling groups for producing and making their model output available.

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

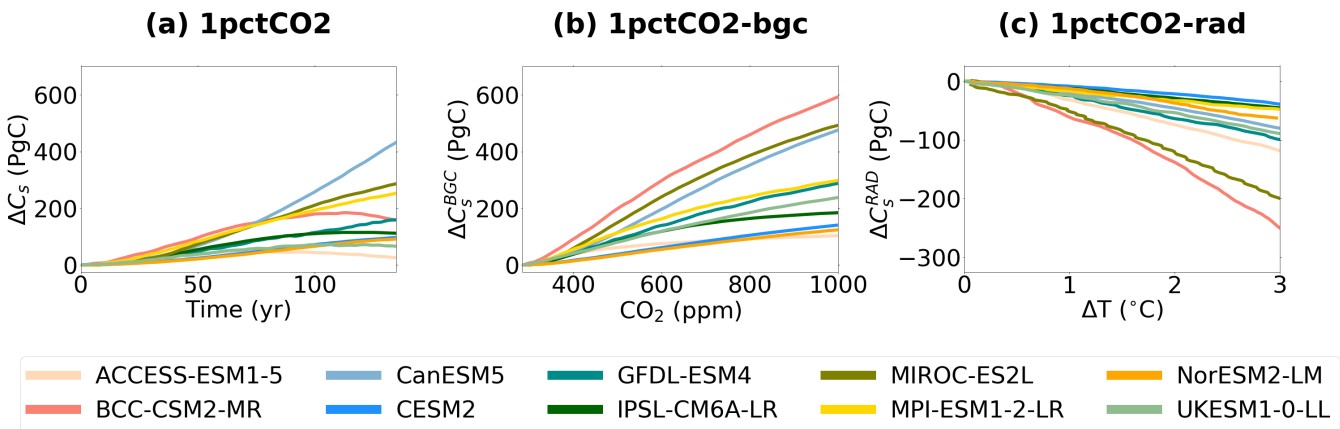

**Figure 1.** Timeseries of projected changes in soil carbon ($\Delta C_s$) in CMIP6 ESMs, for the: (a) idealised 1% $CO_2$ (left column), (b) biogeo-chemically coupled 1% $CO_2$ (BGC, middle column), and (c) radiatively coupled 1% $CO_2$ (RAD, right column) simulations. This figure has been adapted from Fig. A2 in Varney et al. (2023).

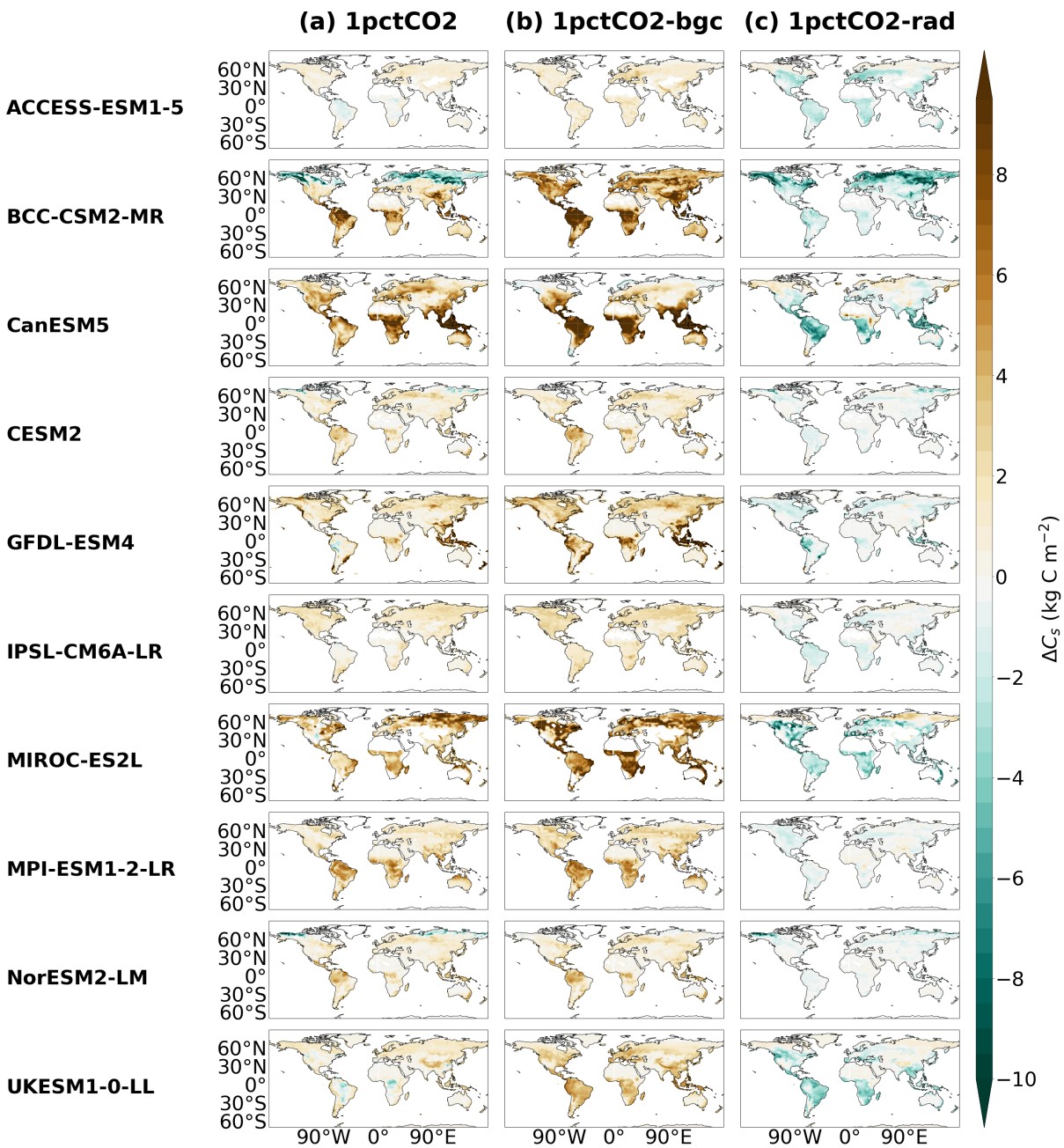

**Figure 2.** Maps showing the changes in soil carbon ($\Delta C_s$) at $4xCO_2$ in CMIP6 ESMs, for the: (a) idealised simulations 1% $CO_2$ (left column), (b) biogeochemically coupled 1% $CO_2$ (BGC, middle column), and (c) radiatively coupled 1% $CO_2$ (RAD, right column).

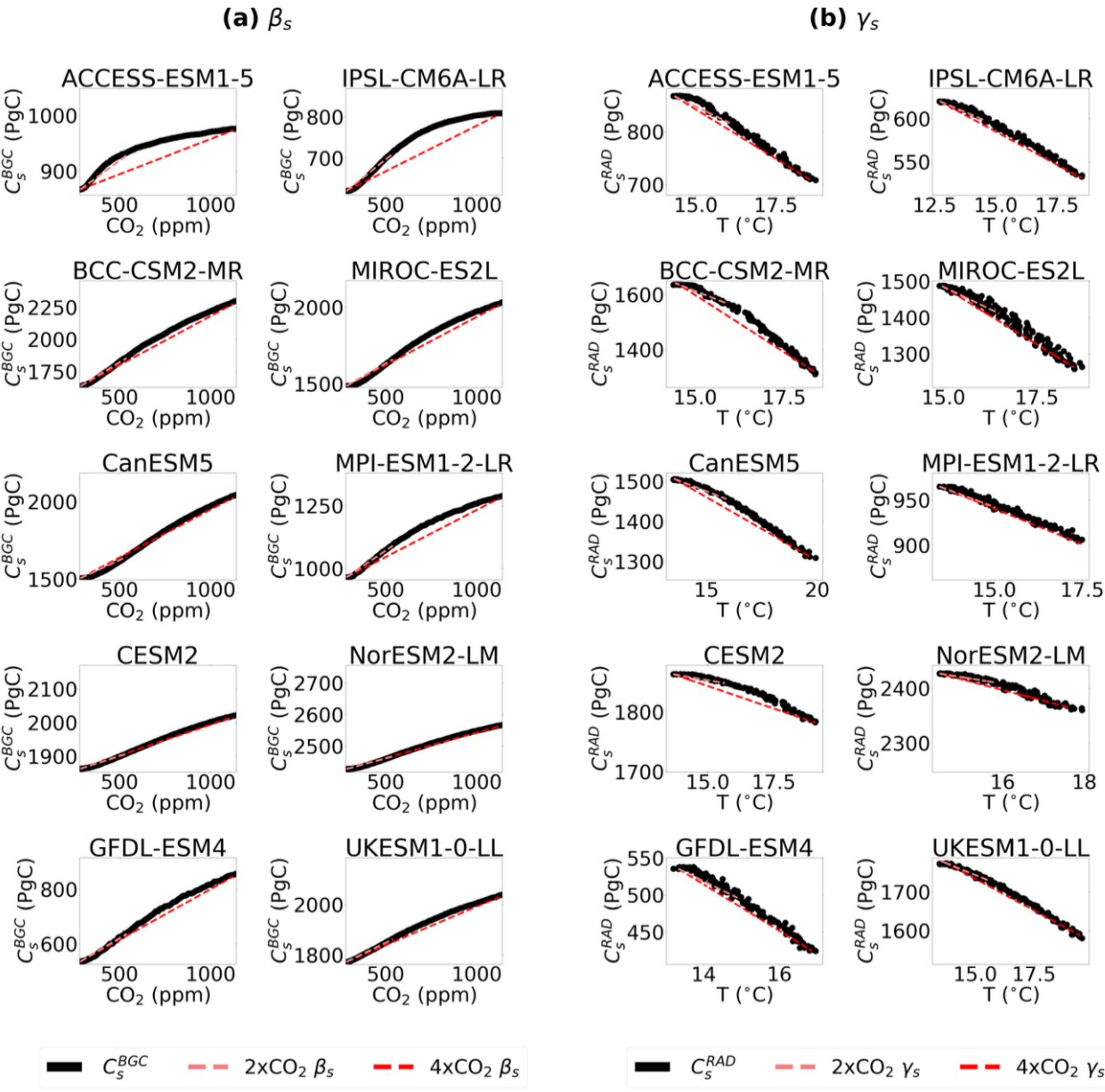

**Figure 3.** Timeseries plots used to calculate the soil feedback parameters. (a) Soil carbon in the BGC simulation ($C_s^{BGC}$, PgC) Vs $CO_2$ (ppm) for the carbon-concentration feedback parameters ($\beta_s$, PgC ppm$^{-1}$), and (b) Soil carbon in the RAD simulation ($C_s^{RAD}$, PgC) Vs temperature (T, °C) for the soil carbon-climate feedback parameters ($\gamma_s$, PgC °C$^{-1}$), for each CMIP6 ESM. The lines show the gradients at 2xCO$_2$ (lighter line) and 4xCO$_2$ (darker line), respectively.

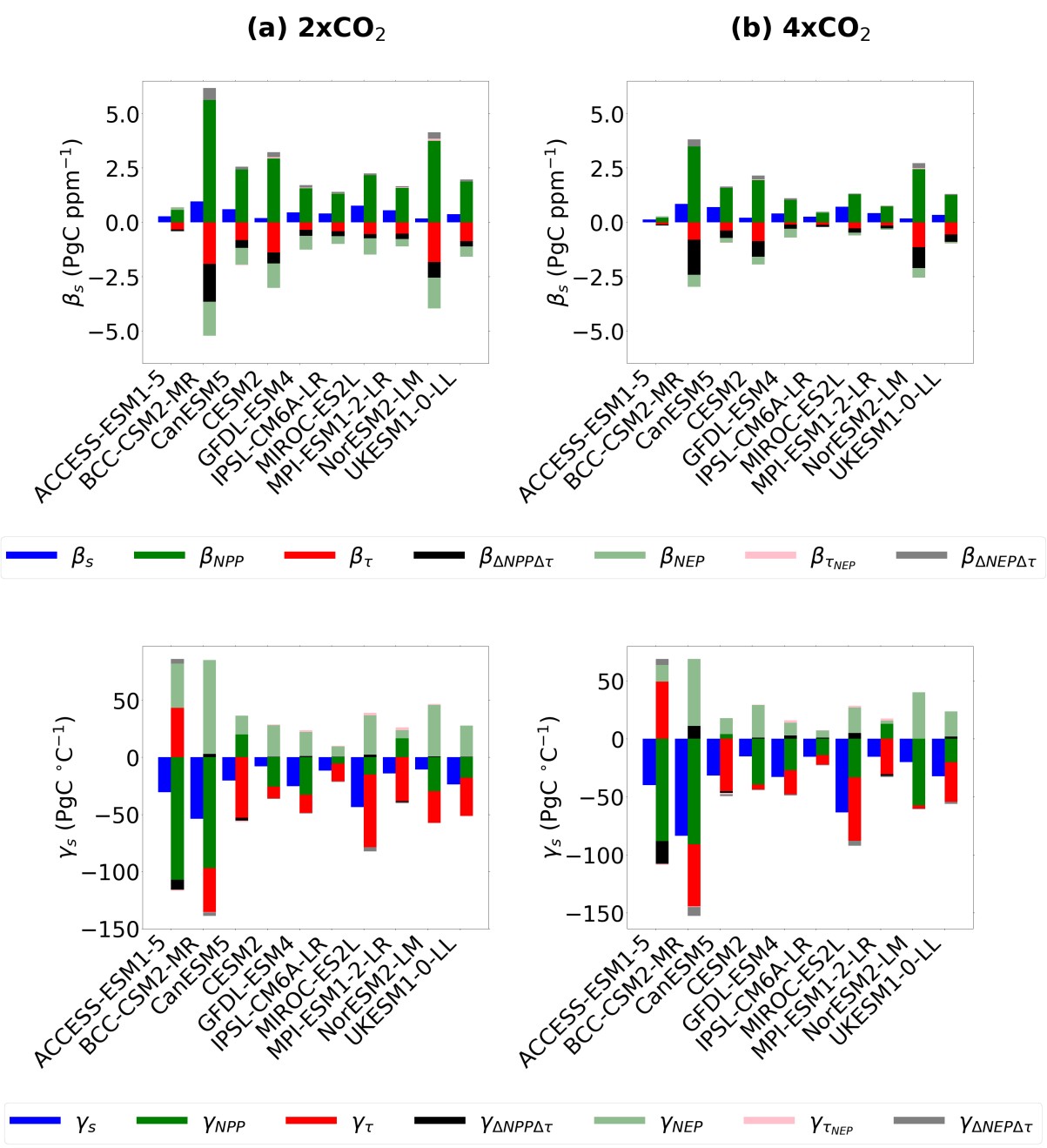

**Figure 4.** Investigating the contribution of individual soil carbon drivers to the soil carbon-concentration ($\beta_s$, top row) and carbon-climate ($\gamma_s$, bottom row) feedback parameters, for each CMIP6 ESM, for (a) 2xCO$_2$ and (b) 4xCO$_2$. The figure shows soil carbon feedback parameter contributions from NPP ($\beta_{NPP}$ and $\gamma_{NPP}$), $\tau_s$ ($\beta_\tau$ and $\gamma_\tau$), the non-linearity in NPP and $\tau_s$ ($\beta_{\Delta NPP\Delta\tau}$ and $\gamma_{\Delta NPP\Delta\tau}$), and the effect from the non-equilibrium term NEP ($\beta_{NEP}$, $\beta_{\tau_{NEP}}$, $\beta_{\Delta NEP\Delta\tau}$ and $\gamma_{NEP}$, $\gamma_{\tau_{NEP}}$, $\gamma_{\Delta NEP\Delta\tau}$).

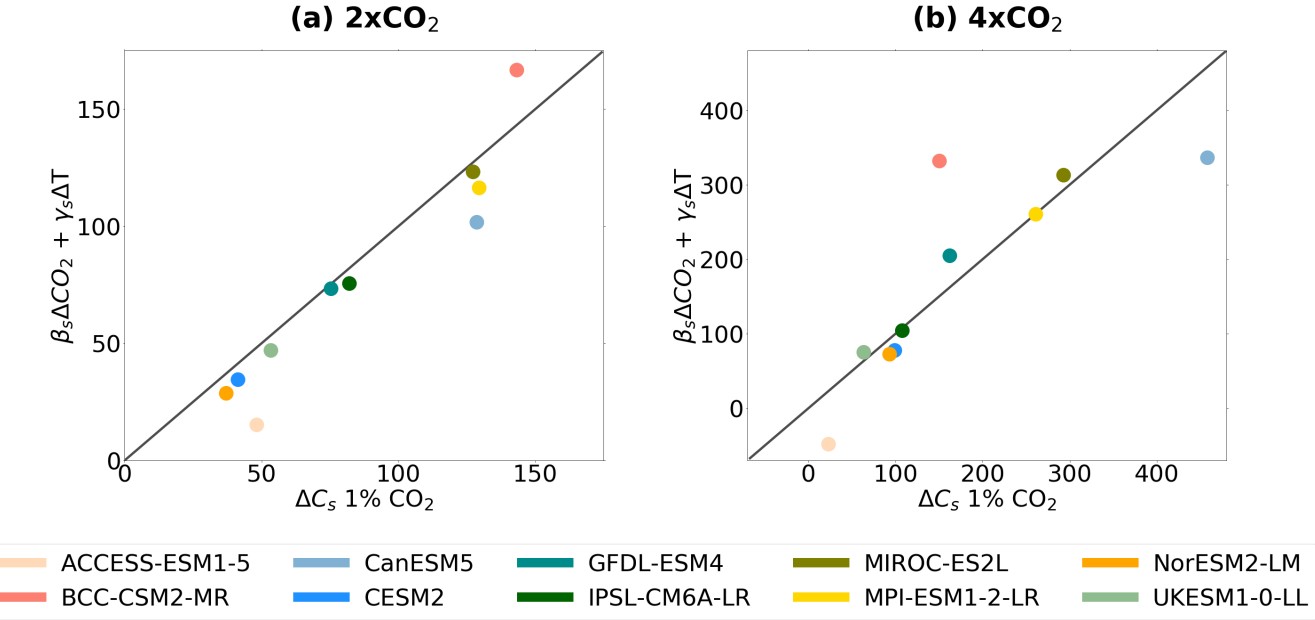

**Figure 5.** Comparison of $\Delta C_s$ (PgC) in the full 1% $CO_2$ simulation (x-axis) against the estimated $\Delta C_s$ using the calculated $\beta_s$ and $\gamma_s$ feedback parameters (y-axis), where estimated $\Delta C_s \approx \beta_s \Delta CO_2 + \gamma_s \Delta T$, for each CMIP6 ESM at (a) 2x$CO_2$ and (b) 4x$CO_2$.

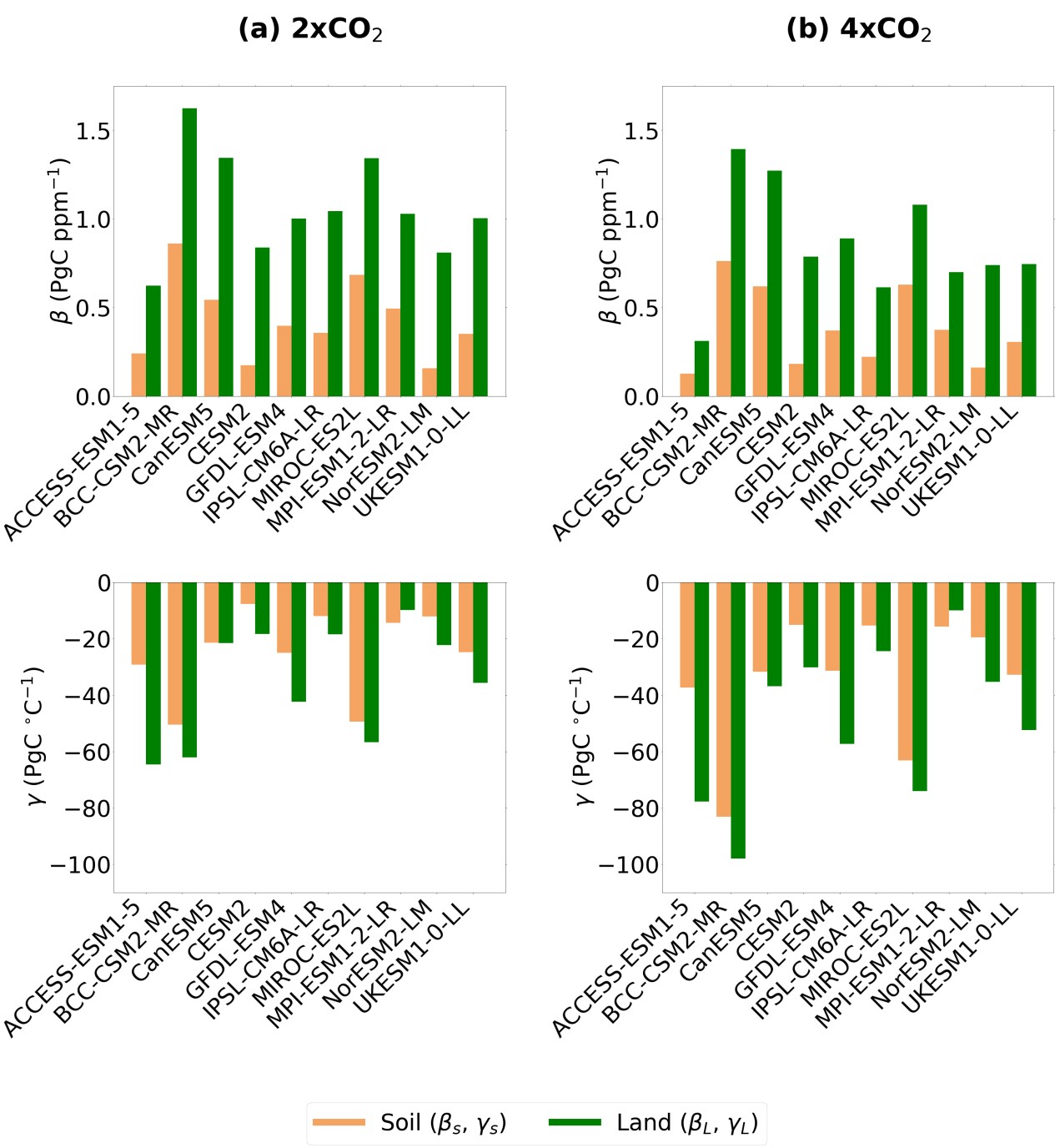

**Figure 6.** Comparison of the land carbon-concentration ($\beta_L$) feedback parameters with the soil carbon-concentration ($\beta_s$) feedback parameters (top row), and the land carbon-climate ($\gamma_L$) feedback parameters with the soil carbon-climate ($\gamma_s$) feedback parameters (bottom row), for (a) 2xCO$_2$ and (b) 4xCO$_2$.

**Table 1.** The CMIP6 Earth system models included in this study and the relevant features of associated land carbon cycle components: simulation of interactive nitrogen, the inclusion of dynamic vegetation, representation of fire, and the soil decomposition functions used (Varney et al., 2022; Arora et al., 2020). Explanations of the temperature and moisture functions used within ESMs are given in Varney et al. (2022) and Todd-Brown et al. (2013).

| Earth System Model | Nitrogen Cycle | Dynamic Vegetation | Fire | Temperature & Moisture Functions |
|---|---|---|---|---|
| ACCESS-ESM1.5 | Yes | No | No | Arrhenius & Hill |
| BCC-CSM2-MR | No | No | No | Hill & Hill |
| CanESM5 | No | No | No | $Q_{10}$ & Hill |
| CESM2 | Yes | No | Yes | Arrhenius & Increasing |
| GFDL-ESM4 | No | Yes | Yes | Hill & Increasing |
| IPSL-CM6A-LR | No | No | No | $Q_{10}$ & Increasing |
| MIROC-ES2L | Yes | No | No | Arrhenius & Increasing |
| MPI-ESM1.2-LR | Yes | Yes | Yes | $Q_{10}$ & Increasing |
| NorESM2-LM | Yes | No | Yes | Arrhenius & Increasing |
| UKESM1-0-LL | Yes | Yes | No | $Q_{10}$ & Hill |

**Table 2.** The soil carbon-concentration ($\beta_s$, PgC ppm$^{-1}$) and carbon-climate ($\gamma_s$, PgC $^\circ$C$^{-1}$) feedback parameters for 2xCO$_2$ and 4xCO$_2$ for the CMIP6 ESMs.

| Earth System Model | 2xCO$_2$ | | 4xCO$_2$ | |
|---|---|---|---|---|
| | $\beta_s$ | $\gamma_s$ | $\beta_s$ | $\gamma_s$ |
| ACCESS-ESM1.5 | 0.242 | -29.2 | 0.127 | -37.3 |
| BCC-CSM2-MR | 0.861 | -50.5 | 0.763 | -83.1 |
| CanESM5 | 0.544 | -21.4 | 0.620 | -31.8 |
| CESM2 | 0.175 | -7.67 | 0.183 | -15.1 |
| GFDL-ESM4 | 0.397 | -25.0 | 0.371 | -31.4 |
| IPSL-CM6A-LR | 0.357 | -11.9 | 0.222 | -15.3 |
| MIROC-ES2L | 0.684 | -49.4 | 0.630 | -63.1 |
| MPI-ESM1-2-LR | 0.494 | -14.4 | 0.375 | -15.6 |
| NorESM2-LM | 0.157 | -12.0 | 0.161 | -19.5 |
| UKESM1-0-LL | 0.351 | -24.7 | 0.307 | -32.7 |
| Ensemble mean | 0.426 | -24.6 | 0.376 | -34.5 |
| Ensemble std | $\pm$ 0.213 | $\pm$ 14.2 | $\pm$ 0.212 | $\pm$ 21.3 |

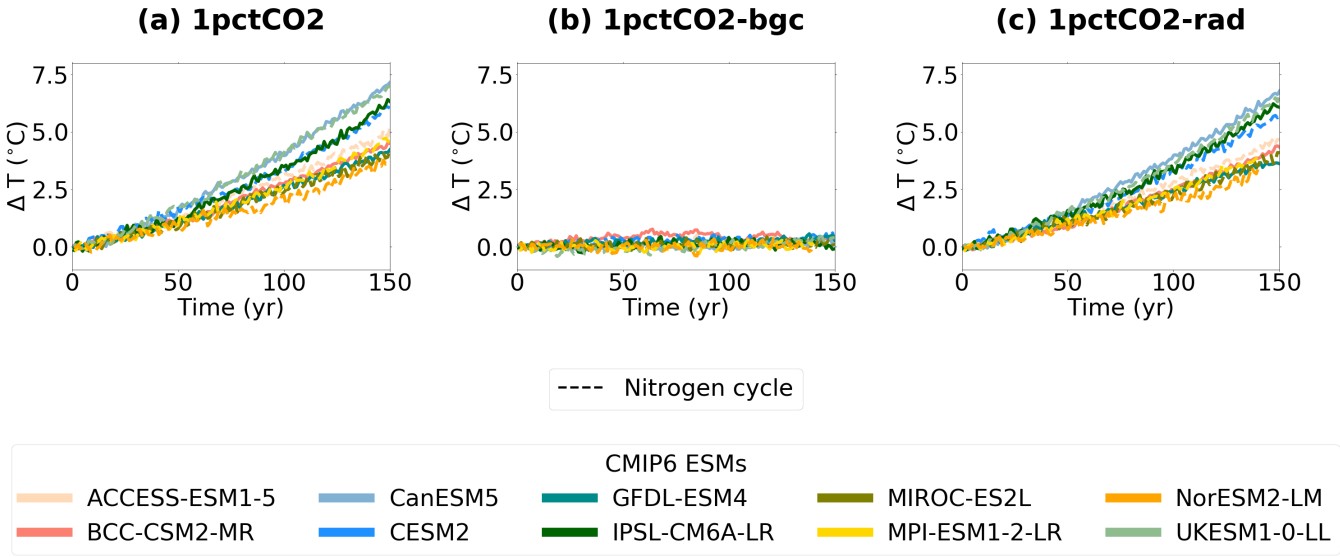

**Figure A1.** Timeseries of projected global mean temperature changes ($\Delta T$) in CMIP6 ESMs for the idealised simulations 1% $CO_2$ (left column), biogeochemically coupled 1% $CO_2$ (BGC, middle column) and radiatively coupled 1% $CO_2$ (RAD, right column).

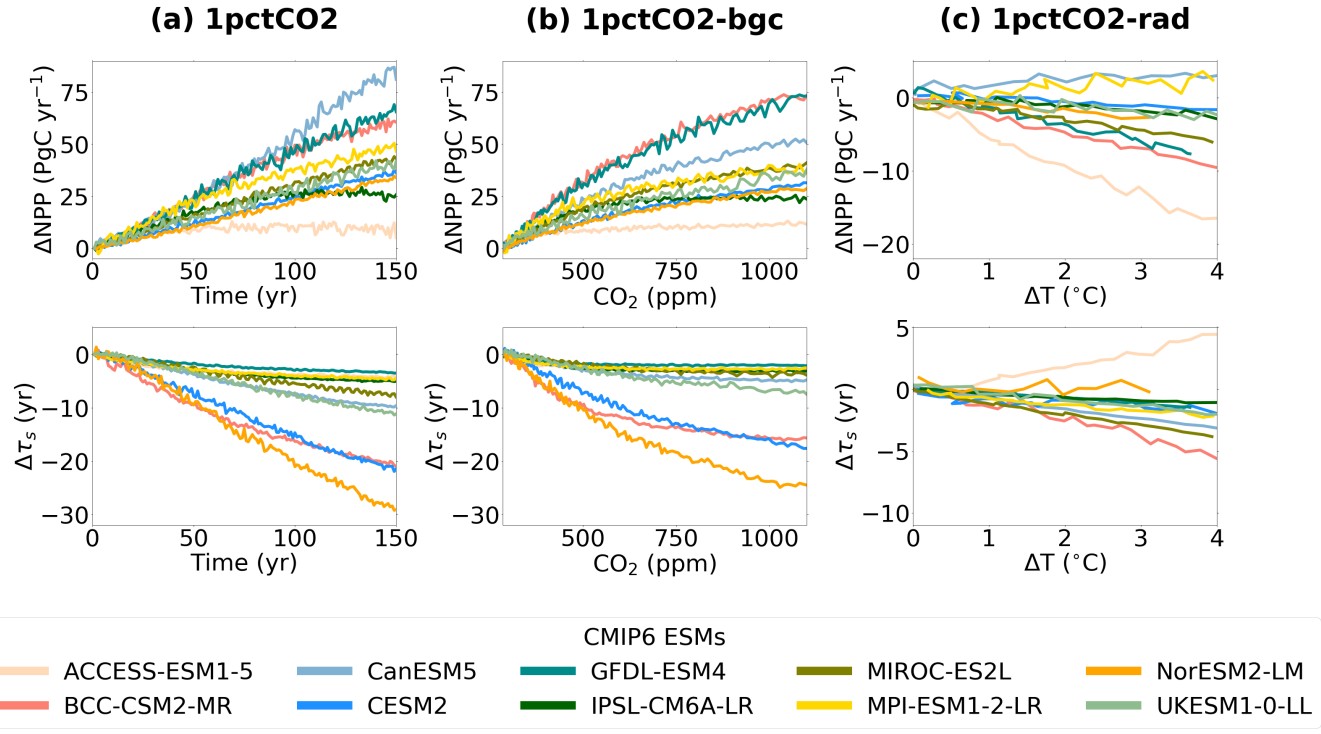

**Figure A2.** Timeseries of projected changes in Net Primary Productivity ($\Delta$NPP, top row) and soil carbon turnover time ($\Delta\tau_s$, bottom row) in CMIP6 ESMs for the idealised simulations 1% $CO_2$ (left column), biogeochemically coupled 1% $CO_2$ (*BGC*, middle column) and radiatively coupled 1% $CO_2$ (*RAD*, right column). This figure has been adapted from Fig. A2 in Varney et al. (2023).

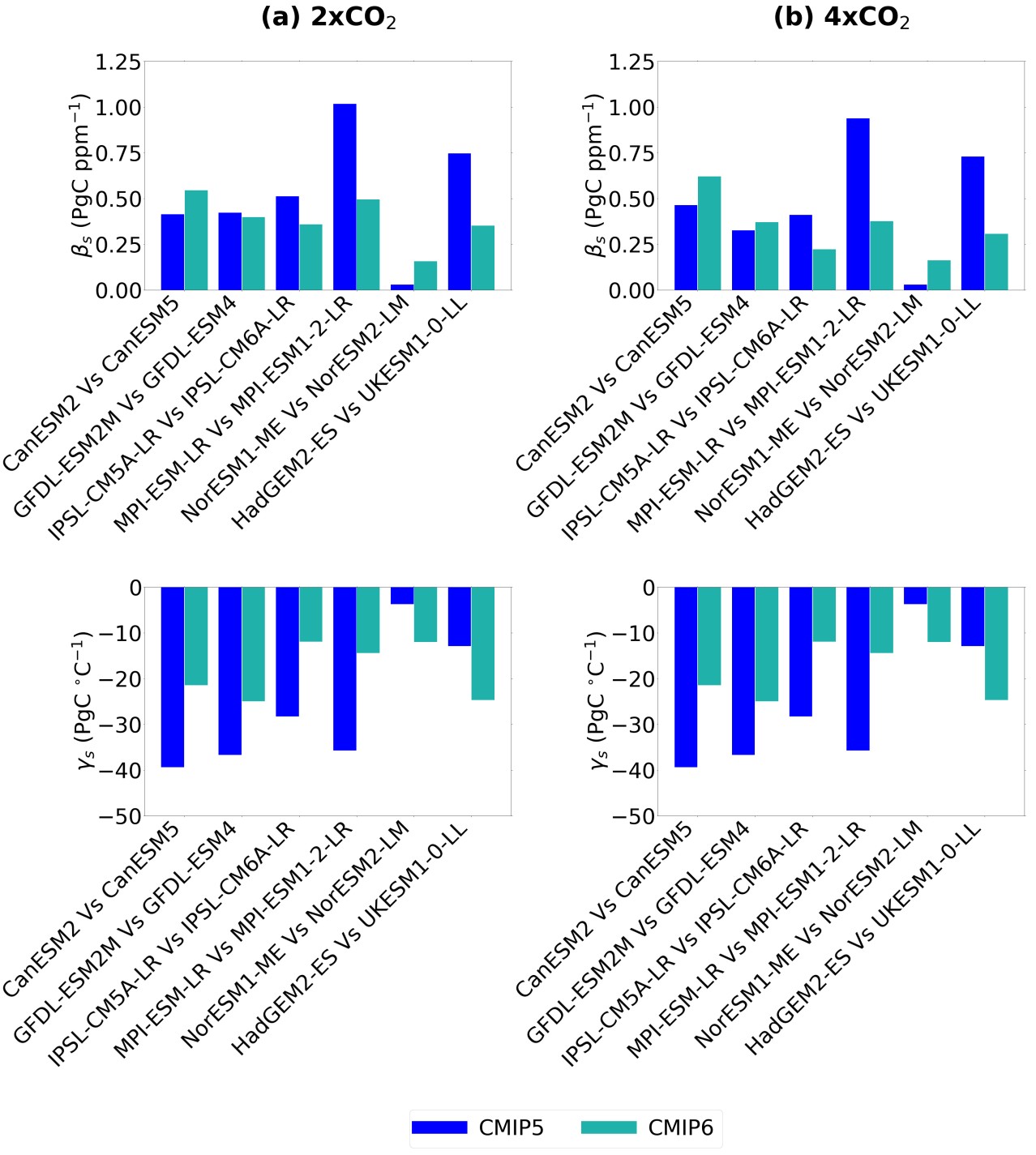

**Figure A3.** Comparison of the soil carbon-concentration ($\beta_s$) feedback parameters (top row) and the soil carbon-climate ($\gamma_s$) feedback parameters (bottom row) from generationally related ESMs from CMIP5 and CMIP6, for (a) 2x$CO_2$ and (b) 4x$CO_2$.

**Table A1.** The land carbon-concentration ($\beta_L$, PgC ppm$^{-1}$) and carbon-climate ($\gamma_L$, PgC $^\circ$C$^{-1}$) feedback parameters for 2xCO$_2$ and 4xCO$_2$ for the CMIP6 ESMs.

| Earth System | 2xCO$_2$ | | 4xCO$_2$ | |
| --- | --- | --- | --- | --- |
| Model | $\beta_L$ | $\gamma_L$ | $\beta_L$ | $\gamma_L$ |
| ACCESS-ESM1.5 | 0.624 | -64.5 | 0.312 | -77.7 |
| BCC-CSM2-MR | 1.63 | -62.1 | 1.39 | -98.0 |
| CanESM5 | 1.34 | -21.6 | 1.27 | -36.9 |
| CESM2 | 0.839 | -18.3 | 0.787 | -30.1 |
| GFDL-ESM4 | 1.00 | -42.3 | 0.891 | -57.3 |
| IPSL-CM6A-LR | 1.05 | -18.4 | 0.614 | -24.5 |
| MIROC-ES2L | 1.34 | -56.7 | 1.08 | -74.0 |
| MPI-ESM1-2-LR | 1.03 | -9.81 | 0.699 | -9.98 |
| NorESM2-LM | 0.811 | -22.2 | 0.740 | -35.3 |
| UKESM1-0-LL | 1.00 | -35.6 | 0.746 | -52.4 |
| Ensemble mean | 1.07 | -35.2 | 0.854 | -49.6 |
| Ensemble std | $\pm$ 0.281 | $\pm$ 19.1 | $\pm$ 0.304 | $\pm$ 26.0 |

**Table A2.** The CMIP5 Earth system models included in this study and the relevant features of associated land carbon cycle components: simulation of interactive nitrogen, the inclusion of dynamic vegetation, and the soil decomposition functions used (Varney et al., 2022; Arora et al., 2013; Anav et al., 2013; Friedlingstein et al., 2014). Explanations of the temperature and moisture functions used within ESMs are given in Varney et al. (2022) and Todd-Brown et al. (2013).

| Earth System Model | Nitrogen Cycle | Dynamic Vegetation | Temperature & Moisture Functions |
|---|---|---|---|
| CanESM2 | No | No | $Q_{10}$ & Hill |
| GFDL-ESM2M | No | Yes | Hill & Increasing |
| IPSL-CM5A-LR | No | No | $Q_{10}$ & Increasing |
| MPI-ESM-LR | No | Yes | $Q_{10}$ & Increasing |
| NorESM1-ME | Yes | No | Arrhenius % Increasing |
| HadGEM2-ES | No | Yes | $Q_{10}$ & Hill |

**Table A3.** The soil carbon-concentration ($\beta_s$, PgC ppm$^{-1}$) and carbon-climate ($\gamma_s$, PgC $^{\circ}$C$^{-1}$) feedback parameters for 2xCO$_2$ and 4xCO$_2$ for the CMIP5 ESMs.

| Earth System Model | 2xCO$_2$ | | 4xCO$_2$ | |
|---|---|---|---|---|
| | $\beta_s$ | $\gamma_s$ | $\beta_s$ | $\gamma_s$ |
| CanESM2 | 0.413 | -39.4 | 0.463 | -54.2 |
| GFDL-ESM2M | 0.421 | -36.7 | 0.326 | -73.5 |
| IPSL-CM5A-LR | 0.511 | -28.3 | 0.410 | -39.5 |
| MPI-ESM-LR | 1.02 | -35.7 | 0.937 | -63.6 |
| NorESM1-ME | 0.0281 | -3.76 | 0.0287 | -7.80 |
| HadGEM2-ES | 0.745 | -12.9 | 0.729 | -18.0 |
| Ensemble mean | 0.522 | -26.1 | 0.482 | -42.8 |
| Ensemble std | ± 0.306 | ± 13.3 | ± 0.290 | ± 23.7 |