# Peer review of "Soil carbon-concentration and carbon-climate feedbacks in CMIP6 Earth system models"

_EGUsphere, 2023_

## Referee Comment (RC2)

**Review for by Soil carbon-concentration and carbon-climate feedbacks in CMIP6 Earth system models by Varney et al., submitted to Biogeosciences (EGUsphere)**

The authors present an analysis of soil carbon cycle feedbacks using CMIP6 models forced with the 1pct-$CO_2$ experiment. Feedbacks are quantified using the integrated flux-based feedback framework from Friedlingstein et al. (2006), referred to here as the $\beta\gamma$ formulation. Feedback parameters are computed from the biogeochemically and radiatively coupled simulations for the carbon-concentration ($\beta$) and carbon-climate ($\gamma$) feedbacks respectively. The study concludes that the sensitivity of soil carbon to climate change increases with warming and is more dominant than the vegetation carbon response, underscoring the importance of soil carbon in long-term land carbon storage. The manuscript is generally clear, and the text has a logical flow through introduction to conclusion. However, I have two main issues:

1. While this study is a useful contribution, it would benefit from an expansion of the analysis. For example, the processes driving soil carbon change in each simulation could and it would be interesting to discuss why the soil carbon response differs between models, although I understand this may require a substantial amount of additional work. I also notice that some sections of the discussion read as a literature review, which could be remedied by better linking the spatial analysis results to driving mechanisms.

2. The manuscript seems to address two kinds of non-linearities: (1) the non-linearity in the soil carbon responses to $CO_2$ and temperature, and (2) another form of non-linearity which arises from non-additivity in the responses in the BGC and RAD simulations to that in the full simulation. In the results section, both non-linearities are mentioned, but in the discussion, it appears that the two are combined and given the same explanation.

**Minor comments**

L3: I suggest replacing the word *feedback* with *response*. The soil carbon responses to $CO_2$ and climate change give rise to the feedbacks.

L5: Please maintain consistency in the terminology used throughout the manuscript. The feedbacks are mostly referred to as soil carbon-concentration feedbacks (without the word *specific*).

L12: Increases in global temperature are an indicator of climate change, not an impact of climate change. Perhaps, rephrase to "sensitivity to climate change and the associated impacts such as changes in precipitation patterns."

L23: Yes, the $\beta\gamma$ formulation can be used for calculating feedbacks from both concentration-driven and emissions-driven simulations, but there are issues with using the latter. Land-ocean compensation due to differing timescales of carbon uptake and loss between the land and ocean affects the magnitude of feedback parameters, so to ensure that both land and ocean see the same atmospheric $CO_2$ concentrations, concentration-driven simulations are used more widely.

L30: Does the first research question also explore the sensitivity of soil carbon to atmospheric $CO_2$? If so, please clarify.

L32: In the third research question, I suggest changing "land surface response" to "land carbon response". This appears to be more consistent with the results presented, which focus on carbon.

L43: Quick clarification on the length of your simulations: are they 140 or 150 years long? L99 refers to 140 years. Please clarify.

L64-66: In equations 2-4, I suggest using the 'approximately equal to' signs between the integrated flux term and the linearization as in Equation 1.

L67-68: The change in atmospheric $CO_2$ concentration is consistent in all three simulations, correct? Omitting the RAD simulation from Line 67 implies otherwise. Please address. With that said, I do understand that the point you want to make here is that the carbon cycle in the RAD simulation sees preindustrial $CO_2$ concentration (no $CO_2$ change) unlike that in the full and BGC simulations.

L76: Accidental *S* added to the Fig. A1 reference in brackets.

L117-120: According to Figure 1, $\Delta C_s$ continues to increase in the GFDL model, whereas in IPSL, the $\Delta C_s$ saturates. The sentence here states the opposite. Perhaps the positions of the two models were switched in this sentence or the labels on the figure were switched. Please review.

L121: $\Delta C_s$ is used to refer to any simulation (full, BGC, or RAD) in the results section. However, in equations 2 - 4, $\Delta C_s$ (with no superscript) is denoted as soil carbon change for the full simulation, and superscripts BGC and RAD are added for the BGC and RAD simulations. It would make the results easier to follow if this is maintained in the results section.

I also suggest using the phrasing "can be approximated by" rather than "is the net effect of" here because responses in the BGC and RAD simulations are not always additive to the response in the full simulation.

L134: Did you mean to say "the *spatial distribution* of $\Delta C_s$ seen in the full 1% simulation …"? Please clarify.

L152: I suggest changing the phrase to "increasing range with increased *global temperature*" because, in the RAD simulation, where $\gamma$ is quantified from, the carbon cycle sees no change in atmospheric $CO_2$.

L154: Figure 3 is confusing. Please explain how the $2xCO_2$ and $4xCO_2$ $\beta$ and $\gamma$ lines were plotted? Could these results be presented differently to improve clarity?

L229: On the other hand, nitrogen mineralization - the temperature-dependent process by which nitrogen in organic matter is converted into inorganic forms that can be taken up by plants – fertilizes soils, countering the limit on productivity.

L259: Missing word *carbon* in "the sensitivity of soil ^ to changes in global temperature"

L262: This could be more concise by phrasing as either "long-term land carbon response under …" OR "long-term land carbon storage under …"

---

## Author Comment (AC1)

Reviewer Comments
Author Responses

**General comments**
This manuscript describes an analysis of CMIP6 outputs, specifically presenting a computation of the soil, vegetation, and land carbon sensitivity to CO2 (beta) and climate (gamma) changes. This follows earlier, similar analyses of previous CMIP generations, and is a useful diagnostic of model behavior and to help understand earth system response to anthropogenic CO2 emissions. The ms is well written and interesting, focused, and generally clear; I applaud the inclusion of a link to the analytical code.
We thank the reviewer for their positive and constructive comments.

In addition to some minor issues, my overall concern is that the analysis is quite limited, misses some very interesting possibilities, and doesn't always display its results well. Specifically, a comparison of CMIP5 and CMIP6 values—ideally quantitatively in a figure, but at least treated in the discussion—would add a lot of value. In addition, the bar graphs are not particularly illuminating, and consider better visualization options (e.g. #9 below).
In summary, this is an interesting and valuable contribution, but needs moderate revisions for concision and clarity; to improve how it conveys its results; and, ideally, to expand its scope a bit.
This comment has been taken on board and addressed in our manuscript. Firstly, we now include a comparison of the CMIP6 soil carbon beta and gamma values with equivalent values from CMIP5 models, which are presented in a new table and new figure (**see below**) within the appendix. The following text has been added to the Results:
'**The βs and γs values were also calculated for CMIP5 ESMs (Table A3), which can be compared with a subset of generationally related CMIP6 ESMs considered in this study (Fig. A3). The CMIP6 ensemble means for both βs and γs parameters are found to be lower compared with the CMIP5 ensemble means (Table A3 and Table 2). The relationship of βs and γs values between CMIP5 and CMIP6 however, is not found to be consistent amongst the ensembles. For βs, reductions are seen in *4 ESMs (GFDL-ESM2M Vs GFDL-ESM4, IPSL-CM5A-LR Vs IPSL-CM6A-LR, MPI-ESM-LR Vs MPI-ESM1-2-LR, and HadGEM2-ES Vs UKESM1-0-LL), compared to increases in the remaining 2 (CanESM2 Vs CanESM5 and NorESM1-ME Vs NorESM2-LM). For γs, a greater value (closer to 0) is seen in 4 ESMs (CanESM2 Vs CanESM5, GFDL-ESM2M Vs GFDL-ESM4, IPSL-CM5A-LR Vs IPSL-CM6A-LR, and MPI-ESM-LR Vs MPI-ESM1-2-LR), compared to a lower value (greater negative) is seen in the remaining 2 ESMs (NorESM1-ME Vs NorESM2-LM and HadGEM2-ES Vs UKESM1-0-LL).***'.

Also, see comments to #9 that Fig. 4 has been updated from the bar chart to the suggested scatter graph (**see Fig. attached**).

Additionally, the analysis has now been expanded to include a breakdown analysis of the processes driving soil carbon in each simulation. The $\beta_s$ and $\gamma_s$ feedback parameters are broken down into sensitivity components due to changes in Net Primary Productivity (NPP) and changes due to soil carbon turnover time ($\tau_s$), which follows the framework presented in Varney et al. 2023 (Biogeosciences). The manuscript will now include a new Methods section '***Processes driving soil carbon change and relation to the βγ formulation***' describing the formulation and how it relates to the $\beta_s$ $\gamma_s$ formulation presented here, and a results

section '***Breakdown of the feedback parameters into soil carbon drivers***', including a new figure (**attached below**).

*Varney, R. M., Chadburn, S. E., Burke, E. J., Jones, S., Wiltshire, A. J., and Cox, P. M.: Simulated responses of soil carbon to climate change in CMIP6 Earth system models: the role of false priming, Biogeosciences, 20, 3767–3790, https://doi.org/10.5194/bg-20-3767-2023, 2023.*

**Specific comments**

1. Line 11: Jones et al. 2018 http://dx.doi.org/10.1029/2018GL079350 might be a good citation here
   This citation will be added to line 11.

2. 37: "has been"
   This sentence will be changed as stated.

3. 52: This is a long time ago! If any models have been added since then, would it be possible to include them on revision? That said, I'm not trying to make a huge amount of new work for the authors
   Sorry for the confusion here, it has been checked multiple times and there are no more CMIP6 models which provide the required data (https://esgf-index1.ceda.ac.uk/projects/cmip6-ceda/).

**4.** 141-142: the results section has a certain amount of restating things that have already been defined/said in the introduction and methods; consider trimming. This is one example
   Thank you for pointing this out, the results section can be trimmed to be more precise.

5. 239: do you mean "explicit" here?
   Indeed. Has been changed to "***explicit***".

6. 256: missing word? "that beta and gamma linearity is a valid assumption"?
   Sentence has been changed to include the word "***linearity***".

7. 264: thanks for the code transparency. Adding a README to this repo would be useful, and I suggest permanently archiving it (i.e., generating a DOI) using Zenodo
   A "README" file has been added to the GitHub repository. If the paper is accepted for publication, we will do as suggested using Zenodo.

8. Figure 2: move to SI? Not sure how useful this is; maps are very small
   We feel the maps are useful to show the patterns of change in the different experiments and across the CMIP6 Earth system models so have kept in the main manuscript, and given that this is an online journal, they can be expanded by the viewer to see greater detail.

9. Consider whether Figure 4 could be re-thought for clarity and impact. For example, what about plotting deltaCs (x) versus beta+gamma (y) with a 1:1 line, coloring points by 2xCO2 or 4x? That might be a better way to visualize for readers
Figure 4 has been remade to follow the suggestion of plotting deltaCs (x) versus beta+gamma (y) with a 1:1 line (**see Figure below**). Though a colour has been used for each ESM (as in Figure 1) so the reader can identify ESMs when comparing to the 1:1 line, therefore a panel is included for 2xCO2 and 4xCO2.

10. You don't need to say "Bar chart", "Maps", etc. in the figure captions. Readers can see what type of plot it is
The figure captions will be changed to avoid unnecessary information, such as '*bar chart*' and '*table*'.

[Figure]

**New Fig. 4 caption**: '*Comparison of $\Delta C_s$ (PgC) in the full 1% $CO_2$ simulation (x-axis) against the estimated $\Delta C_s$ using the calculated $\beta_s$ and $\gamma_s$ feedback parameters (y-axis), where estimated $\Delta C_s \approx \beta_s \Delta CO_2 + \gamma_s \Delta T$, for each CMIP6 ESM at (a) 2xCO2 and (b) 4xCO2.*'.

[Figure]

**(a) 2xCO₂**

**(b) 4xCO₂**

Legend (top): $\beta_s$ · $\beta_{NPP}$ · $\beta_\tau$ · $\beta_{\Delta NPP\Delta\tau}$ · $\beta_{NEP}$ · $\beta_{\tau_{NEP}}$ · $\beta_{\Delta NEP\Delta\tau}$

Legend (bottom): $\gamma_s$ · $\gamma_{NPP}$ · $\gamma_\tau$ · $\gamma_{\Delta NPP\Delta\tau}$ · $\gamma_{NEP}$ · $\gamma_{\tau_{NEP}}$ · $\gamma_{\Delta NEP\Delta\tau}$

**New Fig caption**: '*Investigating the contribution of individual soil carbon drivers to the soil carbon-concentration ($\beta_s$, top row) and carbon-climate ($\gamma_s$, bottom row) feedback parameters, for each CMIP6 ESM, for (a) 2xCO2 and (b) 4xCO2. The figure shows soil carbon feedback parameter contributions from NPP ($\beta_{NPP}$ and $\gamma_{NPP}$), $\tau_s$ ($\beta_\tau$ and $\gamma_\tau$), the non-linearity in NPP and $\tau_s$ ($\beta_{\Delta NPP\Delta\tau}$ and $\gamma_{\Delta NPP\Delta\tau}$), and the effect from the non-equilibrium term NEP ($\beta_{NEP}$, $\beta_{\tau NEP}$, $\beta_{\Delta NEP\Delta\tau}$ and $\gamma_{NEP}$, $\gamma_{\tau NEP}$, $\gamma_{\Delta NEP\Delta\tau}$).*'.

[Figure]

**CMIP5 New Appendix Fig:** 'Comparison of the soil carbon-concentration ($\beta_s$) feedback parameters (top row) and the soil carbon-climate ($\gamma_s$) feedback parameters (bottom row) from generationally related ESMs from CMIP5 and CMIP6, for (a) $2xCO_2$ and (b) $4xCO_2$'.

---

## Author Comment (AC2)

Reviewer Comments
Author Responses

**Review for by Soil carbon-concentration and carbon-climate feedbacks in CMIP6 Earth system models by Varney et al., submitted to Biogeosciences (EGUsphere)**
The authors present an analysis of soil carbon cycle feedbacks using CMIP6 models forced with the 1pct-CO2 experiment. Feedbacks are quantified using the integrated flux-based feedback framework from Friedlingstein et al. (2006), referred to here as the $\beta\gamma$ formulation. Feedback parameters are computed from the biogeochemically and radiatively coupled simulations for the carbon-concentration ($\beta$) and carbon-climate ($\gamma$) feedbacks respectively. The study concludes that the sensitivity of soil carbon to climate change increases with warming and is more dominant than the vegetation carbon response, underscoring the importance of soil carbon in long-term land carbon storage. The manuscript is generally clear, and the text has a logical flow through introduction to conclusion.
We thank the reviewer for their positive and constructive comments.

However, I have two main issues:
1. While this study is a useful contribution, it would benefit from an expansion of the analysis. For example, the processes driving soil carbon change in each simulation could and it would be interesting to discuss why the soil carbon response differs between models, although I understand this may require a substantial amount of additional work. I also notice that some sections of the discussion read as a literature review, which could be remedied by better linking the spatial analysis results to driving mechanisms.
The analysis has now been expanded to include a breakdown analysis of the processes driving soil carbon in each simulation. The $\beta_s$ and $\gamma_s$ feedback parameters are broken down into sensitivity components due to changes in Net Primary Productivity (NPP) and changes due to soil carbon turnover time ($\tau_s$), which follows the framework presented in Varney et al. 2023 (Biogeosciences). The manuscript will now include a new Methods section '**_Processes driving soil carbon change and relation to the βγ formulation_**' describing the formulation and how it relates to the $\beta_s$ $\gamma_s$ formulation presented here, and a results section '**_Breakdown of the feedback parameters into soil carbon drivers_**', including a new figure (**attached below**). Additionally, the discussion has been rewritten and now links to the new results to back up the discussion throughout, so it is now less like a literature review and more of a discussion of the results.

_Varney, R. M., Chadburn, S. E., Burke, E. J., Jones, S., Wiltshire, A. J., and Cox, P. M.: Simulated responses of soil carbon to climate change in CMIP6 Earth system models: the role of false priming, Biogeosciences, 20, 3767–3790, https://doi.org/10.5194/bg-20-3767-2023, 2023._

The manuscript seems to address two kinds of non-linearities: (1) the non-linearity in the soil carbon responses to CO2 and temperature, and (2) another form of non-linearity which arises from non-additivity in the responses in the BGC and RAD simulations to that in the full simulation. In the results section, both non-linearities are mentioned, but in the discussion, it appears that the two are combined and given

the same explanation.

*Agreed. We have added the following text to make this clearer:*

*'The βγ formulation has many benefits in allowing the quantification and comparison of land and soil carbon feedbacks amongst ESMs. **However, one limitation is due to $\Delta C_s$ not being consistently linear with increasing $CO_2$ and temperature (Fig. 3), so the parameter values depend on the point in time which they are calculated (for example, 2x$CO_2$ or 4x$CO_2$). This has been shown to be due to non-linearities in the processes driving soil carbon feedbacks (Fig. 4), such as the discussed saturation of the $CO_2$ fertilisation effect ($β_{NPP}$; Wang et al. (2020)) and additionally a known $Q_{10}$ dependence of heterotrophic (soil) respiration to temperature ($γ_τ$; Zhou et al. 2009)).***

*Non-linearities between  $CO_2$ and T responses are **also** known and have previously been shown within ESMs in the future land carbon responses (Schwinger et al., 2014; Zickfeld et al., 2011; Gregory et al., 2009).'*

**Minor comments**

L3: I suggest replacing the word *feedback* with *response.* The soil carbon responses to CO2 and climate change give rise to the feedbacks.

We agree that soil carbon responds to $CO_2$ and climate changes which then leads to feedbacks on the climate system. However, the use of the term 'feedbacks' when referring to beta and gamma factors, is common in this field and we maintain it here for continuity. We have however changed the sentence to address this comment.

*"This paper quantifies the global soil carbon changes **due** to changes in..."*

L5: Please maintain consistency in the terminology used throughout the manuscript. The feedbacks are mostly referred to as soil carbon-concentration feedbacks (without the word *specific*).

The manuscript has been checked for consistency and now the feedbacks are always referred to as soil feedbacks (without the word specific).

L12: Increases in global temperature are an indicator of climate change, not an impact of climate change. Perhaps, rephrase to "sensitivity to climate change and the associated impacts such as changes in precipitation patterns."

The words "to *climate change*" has been removed here. This means the sentence now refers to global temperature increase as a result of increased atmospheric $CO_2$ concentrations, rather than as a more general impact of climate change.

L23: Yes, the βγ formulation can be used for calculating feedbacks from both concentration-driven and emissions-driven simulations, but there are issues with using the latter. Land-ocean compensation due to differing timescales of carbon uptake and loss between the land and ocean affects the magnitude of feedback parameters, so to ensure that both land and ocean see the same atmospheric CO2 concentrations, concentration-driven simulations are used more widely.

Good point. This sentence has been removed to avoid confusion.

L30: Does the first research question also explore the sensitivity of soil carbon to atmospheric CO2? If so, please clarify.

Changed to avoid confusion:

'… sensitivity of soil carbon to **increased atmospheric $CO_2$ concentrations and associated climate impacts** by …'.

L32: In the third research question, I suggest changing "land surface response" to "land carbon response". This appears to be more consistent with the results presented, which focus on carbon.

Changed as suggested: ' **carbon**'.

L43: Quick clarification on the length of your simulations: are they 140 or 150 years long? L99 refers to 140 years. Please clarify.

C4MIP simulations for the 1% experiments are 150 years long. However, in this case we are considering $2xCO_2$ (approximately 70 years) and $4xCO_2$ (approximately 140 years).

L64-66: In equations 2-4, I suggest using the 'approximately equal to' signs between the integrated flux term and the linearization as in Equation 1.

Equations changed as suggested.

L67-68: The change in atmospheric CO2 concentration is consistent in all three simulations, correct? Omitting the RAD simulation from Line 67 implies otherwise. Please address. With that said, I do understand that the point you want to make here is that the carbon cycle in the RAD simulation sees preindustrial CO2 concentration (no CO2 change) unlike that in the full and BGC simulations.

That is a fair comment, the paragraph will be changed so the distinction between the 3 experiments is clear. The paragraph now starts:

"*In these equations, $\Delta CO_2(t)$ (ppm)* **is consistent between all scenarios. However, within the RAD simulation …**".

L76: Accidental *S* added to the Fig. A1 reference in brackets.

This has been removed.

L117-120: According to Figure 1, ΔCs continues to increase in the GFDL model, whereas in IPSL, the ΔCs saturates. The sentence here states the opposite. Perhaps the positions of the two models were switched in this sentence or the labels on the figure were switched. Please review.

These models have been switched within the text.

L121: ΔCs is used to refer to any simulation (full, BGC, or RAD) in the results section. However, in equations 2 - 4, ΔCs (with no superscript) is denoted as soil carbon change for the full simulation, and superscripts BGC and RAD are added for the BGC and RAD simulations. It would make the results easier to follow if this is maintained in the results section.

The manuscript has been changed to be consistent throughout, where now ΔCs is used for the full simulation and the superscripts BGC and RAD are used for the BGC and RAD simulations respectively.

I also suggest using the phrasing "can be approximated by" rather than "is the net effect of" here because responses in the BGC and RAD simulations are not always additive to the response in the full simulation.

This sentence has been changed as suggested.

L134: Did you mean to say "the *spatial distribution* of ΔCs seen in the full 1% simulation ..."? Please clarify.

Yes, change made as suggested.

L152: I suggest changing the phrase to "increasing range with increased *global temperature*" because, in the RAD simulation, where γ is quantified from, the carbon cycle sees no change in atmospheric CO2.

Sentence changed as suggested.

L154: Figure 3 is confusing. Please explain how the 2xCO2 and 4xCO2 β and γ lines were plotted? Could these results be presented differently to improve clarity?

The figure has been updated to improve the clarity.

L229: On the other hand, nitrogen mineralization - the temperature-dependent process by which nitrogen in organic matter is converted into inorganic forms that can be taken up by plants – fertilizes soils, countering the limit on productivity.

We have added the following to the discussion:

'*However, it is noted that warming within the soil could accelerate nutrient mineralisation, which could result in a liberation of nitrogen due to increased microbial breakdown of plant litter, alleviating the nutrient limitation in plants (Todd-Brown et al. 2014).*'

*Todd-Brown, K. E. O., Randerson, J. T., Hopkins, F., Arora, V., Hajima, T., Jones, C., Shevliakova, E., Tjiputra, J., Volodin, E., Wu, T., Zhang, Q., and Allison, S. D.: Changes in soil organic carbon storage predicted by Earth system models during the 21st century, Biogeosciences, 11, 2341–2356, https://doi.org/10.5194/bg-11-2341-2014, 2014.*

L259: Missing word *carbon* in "the sensitivity of soil ^ to changes in global temperature"

Word "*carbon*" has been added to sentence.

L262: This could be more concise by phrasing as either "long-term land carbon response under ..." OR "long-term land carbon storage under ..."

Changed to "*long-term **land carbon** response under...*".

[Figure]

**New Fig caption**: 'Investigating the contribution of individual soil carbon drivers to the soil carbon-concentration ($\beta_s$, top row) and carbon-climate ($\gamma_s$, bottom row) feedback parameters, for each CMIP6 ESM, for (a) 2xCO2 and (b) 4xCO2. The figure shows soil carbon feedback parameter contributions from NPP ($\beta_{NPP}$ and $\gamma_{NPP}$), $\tau_s$ ($\beta_\tau$ and $\gamma_\tau$), the non-linearity in NPP and $\tau_s$ ($\beta_{\Delta NPP\Delta \tau}$ and $\gamma_{\Delta NPP\Delta \tau}$), and the effect from the non-equilibrium term NEP ($\beta_{NEP}$, $\beta_{\tau NEP}$, $\beta_{\Delta NEP\Delta \tau}$ and $\gamma_{NEP}$, $\gamma_{\tau NEP}$, $\gamma_{\Delta NEP\Delta \tau}$).'.

---

## Author Response (AR2)

Dear Daniel,

Thank you for your comments on our revised manuscript. The final comments from the reviews have now been addressed (see below).

Best wishes,

Rebecca & Co-authors
* * *
**R1**

The authors provide thoughtful and comprehensive responses to all my previous concerns, and the ms has been significantly strengthened in almost every area (methodological clarity, expanded analysis, data visualization, references). I recommend acceptance and congratulate them on a great job!

One minor technical note: re "Code availability" on page 15, it would be better to create a permanent archive and DOI using Zenodo and reference that, rather than a GitHub repository that could be removed in the future.

The code has now been uploaded to Zenodo: https://doi.org/10.5281/zenodo.10927091.

**R2**

Thank you for putting in substantial work to revise the paper and for including an additional framework breaking down contributions to soil beta and gamma parameters. A few suggestions are included below:

1. Line 105 (Equation 10): the tau_s in the first term on the right side of the equation should be initial tau_s (tau_s_zero), correct? That would be required for this term to cancel out with the C_s_zero on the other side of the equation.
This was very well spotted! This has been corrected.

2. Please include somewhere in Section 2.3 what year the initial values of NEP, tau_s and NPP are calculated from. My assumption would be the last year of the spinup or the first year of the 1% per year simulation, though the former may be more ideal.
This is stated within the first paragraph: "decadal time-average at the start of C4MIP simulation", but is now repeated in the section to improve clarity.

3. My remaining questions are on what you refer to as the "transient NEP effect on tau_s" (5th term on the right side of Equation 13). I am struggling with understanding of the significance and meaning of this term here:

a) Please clarify the meaning of this term further. What is meant by "transient NEP effect on tau_s"? I referred back to the Varney et al. (2023) paper and still find this confusing.

b) Since the 1% per year simulation is initialized from equilibrium, would NPP-R_h, and hence, NEP_zero, not be close to 0? If that is the case, then the "transient NEP effect on tau_s" (5th term on the right side of Equation 13) would be 0 and could be excluded from the rest of the equations. It also appears from Figure 4 that this term makes very little contribution and is, therefore, hardly addressed in the manuscript anyway.

I understand the confusion here as this term is negligible and could have been neglected. It was included for mathematical completeness of the equations, meaning that both sides of the equation are exactly equal to one another. I have included an additional paragraph.

"*The NEP term is used to represent the transient state of the system where NPP does not equal $R_h$. However, it is noted that if the initial state is in equilibrium, then the initial NEP ($NEP_0$) will be approximately equal to zero. This means the $\Delta C_{s,\tau NEP}$ term (representing the difference in $\tau_s$ from using NPP or $R_h$ in the definition) will be negligible. Despite initialising at the start of the C4MIP simulations (decadal time-average at the start of C4MIP simulation), this term is included within the analysis for completeness to ensure exact values of $\Delta C_s$.*".